# Development of a regional probabilistic seismic hazard model for Central Asia

Valerio Poggi[1], Stefano Parolai[2,1], Natalya Silacheva[3], Anatoly Ischuk[4], Kanatbek Abdrakhmatov[5], Zainalobudin Kobuliev[6], Vakhitkhan Ismailov[7], Roman Ibragimov[7], Japar Karaev[8], Paola Ceresa[9], Marco Santulin[1], Paolo Bazzurro[9,10].

[1]National Institute of Oceanography and Applied Geophysics (OGS), Udine, Italy
[2]Department of Mathematics, Informatics and Geosciences, University of Trieste, Trieste, Italy
[3]Institute of Seismology, Ministry of Emergency Situations (MoES) of the Republic of Kazakhstan, Almaty, Kazakhstan
[4]Institute of Geology, Earthquake Engineering and Seismology, National Academy of Sciences of Tajikistan, Dushanbe, Tajikistan
[5]Institute of Seismology, National Academy of Sciences of the Kyrgyz Republic, Bishkek, Kyrgyz Republic
[6]Institute of Water Problems, Hydropower and Ecology (IWPHE), Academy of Sciences of the Republic of Tajikistan, Dushanbe, Tajikistan
[7]Institute of Seismology of the Academy of Sciences of the Republic of Uzbekistan, Tashkent, Uzbekistan
[8]United Nations Development Programme (UNDP) Representative Office in Turkmenistan, Ashgabat, Turkmenistan
[9]Risk Engineering + Development (RED), Pavia Italy
[10]University School for Advanced Studies (IUSS), Pavia, Italy

*Correspondence to*: Valerio Poggi (vpoggi@ogs.it)

**Abstract.** Central Asia is an area characterized by complex tectonic and active deformation, largely due to the relative convergent motion between India and Arabia with Eurasia. The resulting compressional tectonic regime is responsible for the development of significant seismic activity, which, along with other natural hazards such as mass movements and river flooding, contributes to increased risk to local populations. Although several studies have been conducted on individual perils at the local and at national levels, the last published regional model for the whole Central Asia, developed under the EMCA project ("Earthquake Model of Central Asia"), is almost ten years old.

With the goal of developing a new comprehensive multi-risk model, that is uniform and consistent across the five Central Asian countries of Kazakhstan, the Kyrgyz Republic, Tajikistan, Turkmenistan, and Uzbekistan, the European Union, in collaboration with the World Bank and the Global Facility for Disaster Reduction and Recovery (GFDRR), funded the regional program SFRARR ("Strengthening Financial Resilience and Accelerating Risk Reduction in Central Asia"). The activity was led by a consortium of scientists from international research institutions, from both the public and private sectors, with contribution from experts of the local scientific community.

This study presents the main results of a probabilistic seismic hazard analysis (PSHA) conducted as part of the SFRARR program to develop the new risk model for Central Asia. The proposed PSHA model was developed using state-of-the-art methods and calibrated based on the most up-to-date information available for the region, including a novel homogenized earthquake catalogue compiled from global and local sources and a database of active faults with associated slip rate information.

# 1 Introduction

Due to the ongoing collision between India and Arabia with Eurasia, resulting in significant stress accumulation in the Earth's crust around the main tectonic suture zones and up to hundreds of kilometres away (Tunini et al., 2017), Central Asia countries are inherently exposed to high levels of seismicity. Several damaging earthquakes have been reported in recent and historical times (see Poggi et al. 2024 for a comprehensive summary), while the seismic risk is exacerbated by the high vulnerability of the local building stock and infrastructures. Reliable risk assessment is therefore an essential step in developing an effective risk mitigation strategy and forms the basis for the formulation and enforcement of national seismic regulations. However, a reliable seismic risk assessment must be based on an updated and reliable seismic hazard model for the region.

Earthquake hazard in Central Asia has been comprehensively assessed in several national and international studies. A first attempt at regional homogenization was made by the Global Seismic Hazard Assessment Program (GSHAP, Giardini et al., 1999), which aimed to establish a common framework for the uniform assessment of the seismic hazard at global scale. Within this framework, a new seismic zonation for the Central Asia was proposed (Ulomov et al., 1999), a first probabilistic seismic hazard model in macroseismic intensity was established. In 2012, the EMCA ("Earthquake Model of Central Asia") project aimed to develop a new comprehensive seismic hazard and risk model for Central Asia as part of the global earthquake hazard and risk model under development at the Global Earthquake Model (GEM) Foundation. Several datasets have been compiled and published, including a homogenized seismic catalogue and a new earthquake source zonation model. The results of the project have been documented in several publications, such as Bindi et al. (2011, 2012) and Ullah et al. (2015).

Several studies at national level followed the aforementioned regional project EMCA, as presented in Ischuk et al. (2014, 2018) for Kyrgyzstan, Tajikistan and eastern Uzbekistan, Silacheva et al. (2018), and Mosca et al. (2019) for Kazakhstan. A probabilistic earthquake hazard analysis for Kyrgyzstan was performed by Abdrakhmatov et al. (2003), in terms of both peak ground acceleration and Arias Intensity (AI), followed by a more comprehensive model developed under the Central Asia Seismic Risk Initiative (CASRI, Abdrakhmatov, 2009) that also includes fault traces. Studies on seismic hazard of Uzbekistan have been conducted within national programs, e.g., in Abdullabekov et al. (2002, 2012), Artikov et al. (2018, 2020). In addition, seismic hazard studies were conducted in Turkmenistan by the Institute of Seismology and Atmospheric Physics of the Academy of Sciences in the framework of regulatory acts (see Ministry of construction of Turkmenistan, 2017). In 2013, the Ministry of Education and Science of the Republic of Kazakhstan requested the development of probabilistic maps for the general seismic zoning of the Republic of Kazakhstan and the seismic microzoning of Almaty. The maps were developed by the Institute of Seismology of Kazakhstan with the participation of other relevant institutions and are in the process of being implemented in the building code that will guide future construction practices. A package of maps of general seismic zoning is then included in the national Code of Rules No 2.03-30-2017 "Construction in Seismic Zones". In 2020, the Kazakhstan Research Institute of Construction and Architecture began drafting regulatory documents based on the Almaty microzoning map package.

The Institute of Geology, Earthquake Engineering, and Seismology of the National Academy of Sciences of Tajikistan, on behalf of the Government of Tajikistan and with technical assistance from the World Bank, prepared a new probabilistic

seismic hazard map of the territory of Tajikistan in 2020. The results are currently being reviewed by the Committee on Construction and Architecture of the Government of Tajikistan for inclusion in the National Building Code.

The Institute of Seismology NAS KR, Bishkek, Kyrgyzstan (IS), the Institute of Geophysical Research NNC RK, Kazakhstan, the Seismological Experimental Methodological Expedition MES RK, Almaty, Kazakhstan (SEME), the Kazakh National Data Center (KNDC), and the Institute of Geology, Earthquake Engineering and Seismology of the Academy of Sciences of

75 the Republic of Tajikistan (IGEES) were currently participating in the recent ISTC project "Central Asia Seismic Hazard Assessment and Bulletin Unification" (CASHA-BU) (2018-2021). Recently, the President of Uzbekistan signed a new law on "Ensuring Earthquake Safety of the Population and Territory of the Republic of Uzbekistan," which mandates the use of modern approaches to earthquake hazard assessment with the goal of reducing the associated risk to structures and the population.

Today, the availability of new data, local and regional seismotectonic studies, and recently developed methods and tools leads us to develop a new probabilistic seismic hazard model that summarizes the current state of knowledge in Central Asia. With the goal of improving financial resilience and risk-based investment planning, the European Union, in collaboration with the World Bank and GFDRR, has launched the "Strengthening Financial Resilience and Accelerating Risk Reduction in Central Asia" (SFRARR) program to improve disaster and climate resilience in Central Asian countries, which include Kazakhstan,

the Kyrgyz Republic, Tajikistan, Turkmenistan, and Uzbekistan. The program includes several operational components, all of which contribute to the development of a new comprehensive probabilistic risk assessment covering multiple hazards and asset types in the target countries.

The project was led by an international consortium of private and public research organisations, including representatives from Central Asian countries. Some of these representatives had also participated in the previously mentioned national initiatives.

The consortium integrated the experience and feedback of these national experts into the presented model. This feedback significantly influenced numerous modelling decisions, including source zonation, data harmonisation, tectonic regionalisation, and more. In contrast to the individual efforts of previous national initiatives, the SFRARR initiative aimed to harmonise these contributions into a single regional model. This approach was intended to improve and build on previous efforts, such as the EMCA model published in 2012, by bringing together diverse expertise and data sources into a

comprehensive and coherent framework.

In this paper, we describe the implementation of a probabilistic seismic hazard model for Central Asia, developed with the contributions and resources of local scientists primarily involved in the World Bank-funded initiative.

## 2 Methodology

In this study, the seismic hazard of five Central Asian countries (Kazakhstan, Kyrgyzstan, Tajikistan, Turkmenistan, and Uzbekistan) is assessed using a probabilistic approach (e.g., Cornell 1968; McGuire 2004) as formalized in Field et al. (2003). Probabilistic seismic hazard assessment (PSHA) allows estimating the annual probability of exceeding ground motion levels at a site due to events that may be caused by different earthquake sources, each with defined characteristics and seismogenic potential. More specifically, the assessment is made at any given observation site in the study region by evaluating the ground motion level (for a number of different ground motion intensity measures) that has a certain probability of being exceeded within a given observation period (e.g., 50 years). In the simplest representation, each source is considered independent of the others and the earthquake onset process is assumed to follow a Poisson process. Each source is fully described by the geometrical properties (dimension, location, orientation) of all possible ruptures and by the definition of their corresponding temporal occurrence behaviour. While the former requirements can be obtained directly by analysing available earthquake records (e.g., moment tensor solutions) and from geologic and tectonic considerations, the latter requirements must be calibrated from past observed seismicity and using a sufficiently comprehensive earthquake catalogue.

The methodology chosen for the construction of the earthquake source model for the Central Asian countries follows a classical approach, largely based on the analysis of the most recent and up-to-date geological and tectonic information from the scientific literature and on the available earthquake records from global bulletins and local earthquake catalogues.

The developed seismic source model consists of a combination of distributed seismicity (homogeneous area sources and gridded, smoothed rates) and finite faults, the former calibrated on the analysis of the occurrence of a regionally harmonized earthquake catalogue homogenized in the moment magnitude (Mw) scale, while the latter derived from a thorough evaluation of direct geological information from databases of active faults and scientific literature (see Poggi et al., 2024 for a comprehensive description of the input datasets assembled for this regional study). The advantage of such a hybrid source model is a more realistic representation of the spatial pattern of seismicity, which is difficult to reproduce just with standard (homogeneous) area sources (Woessner et al. 2015, Poggi et al., 2020). This approach is particularly valuable when the delineation of higher resolution source areas proves difficult due to limited seismogenic constraints or, alternatively, when very large regions are considered (e.g., Stirling et al., 2012; Moschetti et al. 2015), provided that the local process of earthquake generation is sufficiently understood.

The following sections detail the various components of the Central Asian hazard model, including the seismicity analysis (estimation of occurrence rates, maximum magnitude, definition of dominant faulting style, etc.), and the implementation of the earthquake source model. Other sections are devoted to the regional selection of the most appropriate models for ground motion prediction and the treatment of epistemic uncertainties using a logic tree approach. The seismic hazard was calculated using the OpenQuake engine (Pagani et al., 2014), an open-source seismic hazard and risk calculation software developed, maintained, and distributed by the Global Earthquake Model (GEM) Foundation. In the next sections, the main results and products of the Central Asian model are presented.

## 3 The homogeneous area source model

Discretization of the study area into multiple zones of supposedly uniform temporal and spatial occurrence of earthquakes is the basis of the distributed seismicity approach, in which observed seismicity is not associated with a known (or inferred) tectonic structure but is assumed to occur everywhere in the area with equal probability. In addition, the division into discrete zones is also an essential prerequisite for the calibration of the analytical occurrence model, whose parameters must be constrained by a sufficiently large number of events to ensure statistical significance.

In this study, the homogeneous area source model was implemented primarily on the basis of the harmonized earthquake catalogue for the region (Poggi et al., 2024), taking also into account all the information available for the target region from the scientific literature and previous studies, including geological and seismotectonic interpretations (e.g., description of fault systems and their relationship to local stress and deformation regimes), existing seismicity analyses, and previous seismic hazard assessments from past regional projects (e.g., GSHAP, Giardini et al., 1999, and EMCA) and published studies (e.g., Abdullabekov et al., 2012; Ischuk et al., 2018; Silacheva et al., 2018). The geometry of source areas was defined according to the guidelines proposed by Vilanova et al. (2014), which provide a set of objective criteria for delineating regions of putative homogeneous seismic potential.

The objective of the modelling process was to achieve a well-founded consensus among the consortium participants. It is noteworthy that Central Asian scientific partners and local representatives from several countries actively participated in both the development and review of the model. Accordingly, in light of the different scientific perspectives on potentially contentious matters, it was imperative to achieve a degree of consensus. Local experts provided feedback, which was incorporated into subsequent revisions of the model. These revisions were developed through numerous meetings, topical workshops, and individual communications. During this iterative process, the initial zonation model was shared with partners, and suggestions for revisions were collected and incorporated. The current iteration of the initial model is referred to as "version 6." Due to space limitations, it is not possible to include an exhaustive description of the entire argument supporting the construction of the zonation model. However, an overview of the main aspects is provided below.

In the developed model, three independent layers of zonation have been implemented according to source depth: the standard zonation model for shallow seismicity (< 50km), and two additional layers of zones for intermediate (50-170km) and deep (> 170km) seismicity.

### 3.1 Shallow seismicity zones

The shallow seismicity model describes the earthquake sources down to a depth of 50 km. It consists of 61 homogeneous seismic source areas categorised into seven primary tectonic groups (A to G, **Figure 1**). These groups are assumed to have similar seismic characteristics, particularly in terms of earthquake productivity (quantified by the b-value of the Gutenberg-Richter relation) and rupture mechanisms. These characteristics are inextricably linked to the different rheological properties and stress/deformation regimes of the crust. Comprehensive statistical analyses of the distributions of focal mechanisms and

empirical magnitude frequency distributions were performed (see following section), complemented by an investigation of the main active fault systems (detailed in the companion paper to this study, Poggi et al. 2024), in the context of regional tectonic
structures and boundaries.

To provide a practical yet concise example of the construction process, Zone D has been identified as encompassing a tectonic domain clearly separated by the stable features of the West Siberian craton (Zone E). As indicated by the available source mechanisms, Zone D is characterised by a mixed regime, but dominated by large transpressive fault systems (e.g. Talas-Fergana Fault, Irtysh Shear Zone) that have influenced the southeastern evolution of the Tianshan Massif (Chen et al., 2022).
Towards the south, a change in seismotectonic style becomes evident in Zone G, where reverse mechanisms increasingly dominate and large overthrust systems develop along the suture zone with the former cratonic terrains of the Tarim region (Angiolini et al., 2013). Here, seismic productivity is increasing, and large magnitudes have occurred in the past. Further south, a mixed tectonic style occurs again in Zone C, while in Zone F, seismicity typical of continental collision is observed with larger and deeper events along the Pamir thrust system (e.g. Murodov, 2022). Towards the west, along the ideal southwestern
extent of the Pamir suture zone, a clear distinction between the tectonic styles of the systems at the boundary between the Turan platform (Zone B) and the Karakum terrains (Zone A) has also been observed (see Ghassemi and Garzanti, 2018, for a comprehensive review).

Consistent with the boundaries of the area studied (see the buffer region used to create the harmonized catalogue, Poggi et al., 2024), source zones were drawn within 300 km of the boundaries of the target states.

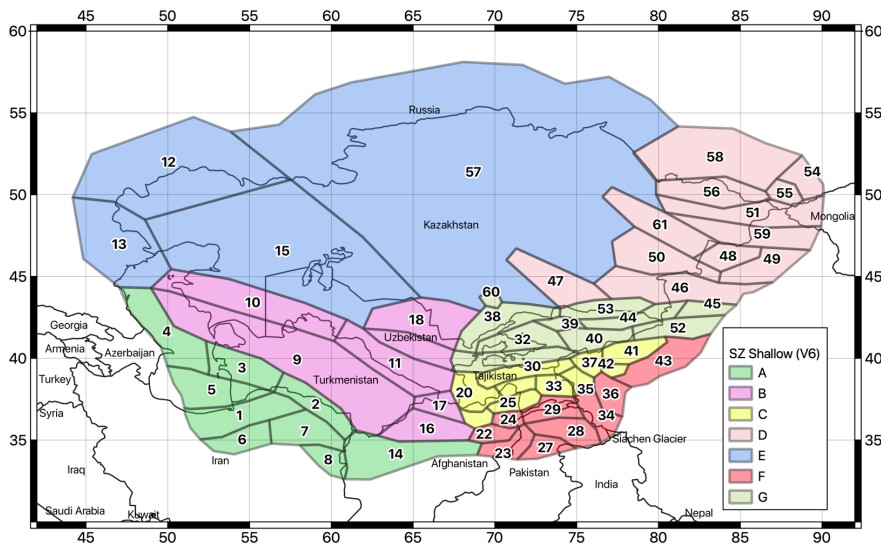

**Figure 1.** Earthquake source zonation model for the shallow crust (<50km). Different colours indicate the different tectonic groups (A to G).

## 3.2 Deep seismicity zones

Analysis of hypocentral depth distribution (see the following section) revealed that a significant proportion of earthquakes occur at depths greater than 40-50 km, which is considered the lowest thickness of continental (brittle) crust in the area. These deep events are clustered into two main regions (**Figure 2**) where there is likely crustal thickening due to the development of deep overthrusts resulting from continental collision. Earthquake sources at these depths have different characteristics from the observed shallow seismicity and should therefore be treated separately. For this reason, two source areas at intermediate depths (H and K) and one at deep depths (L) were implemented separately to represent the seismogenic range of 50-170 km and 170-400 km, respectively.

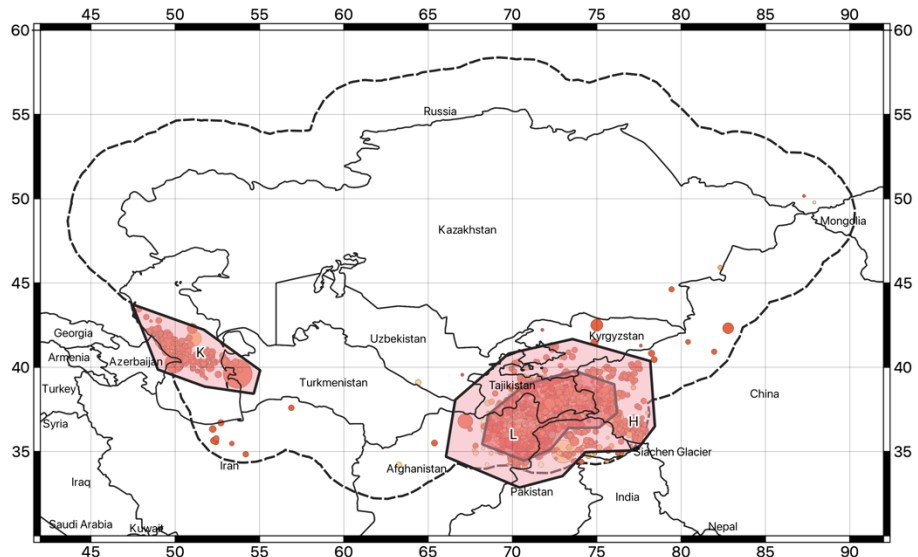

**Figure 2.** Earthquake source zonation model for the intermediate (H and K zones, 50-170km) and deep (L zone, >170km) earthquakes.

## 4 Seismicity analysis

While seismic zonation provides a means to distinguish between regions of different seismic behaviour, the different source properties (e.g., hypocentral depth distribution, temporal occurrence model and dominant rupture mechanism) must then be defined separately for each discrete zone to create the final source model. In the following, a comprehensive description of the source model parameterization is given.

### 4.1 Hypocentral depth distribution

From the analysis of the harmonized earthquake catalogue available for the region, a probability density distribution for depth was estimated for the different source groups (**Figure 3**). Events of unknown depth were excluded from the analysis, as were

events with typical fixed depth solutions (e.g., 0, 5, 10, 33 km, etc.) to avoid biased statistics. Nevertheless, sufficient samples were available to perform a reasonably robust analysis for each source group, allowing the definition of discrete depth distributions consistent with the seismotectonic features expected for the area. Although the input data often lack specific uncertainties related to each individual depth solution, we addressed this by regularizing the probability distributions. This was achieved by applying a smoothing filter, which helps to minimize the impact of single outliers and emphasize the overall trend of variation. The smoothing process allows us to better capture the underlying patterns in earthquake depth distribution, thereby accounting for possible uncertainties and providing a more robust basis for the construction of the seismic hazard model.

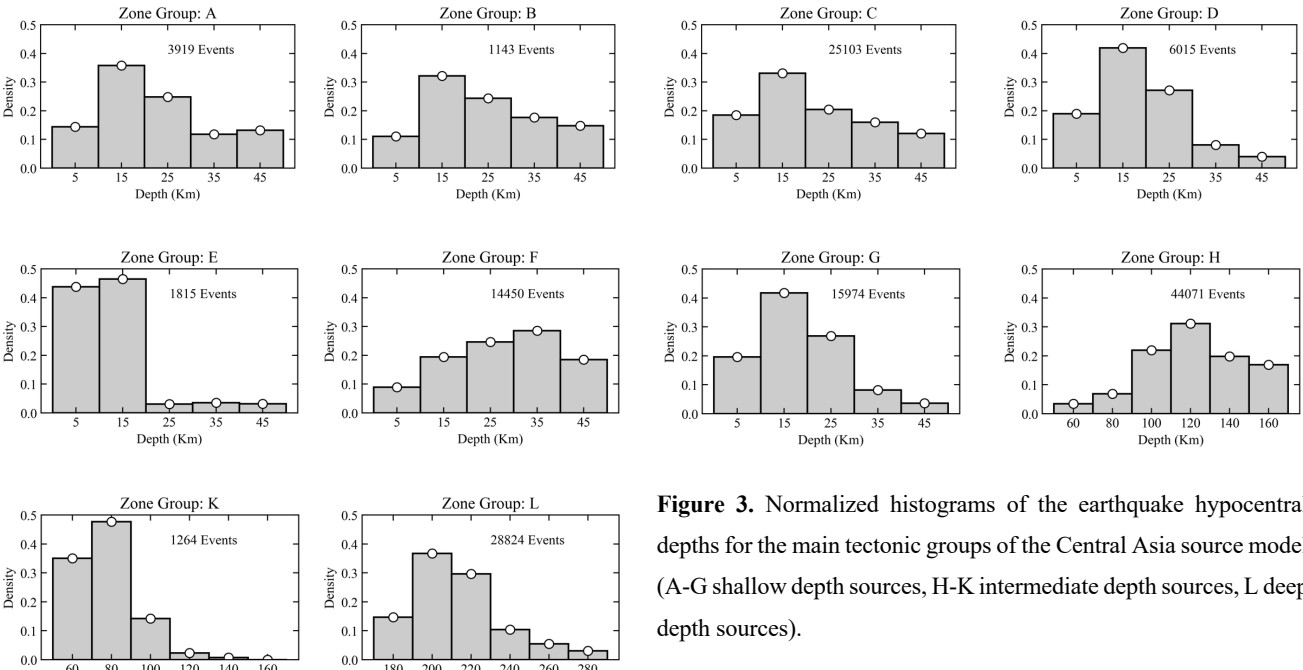

**Figure 3.** Normalized histograms of the earthquake hypocentral depths for the main tectonic groups of the Central Asia source model (A-G shallow depth sources, H-K intermediate depth sources, L deep depth sources).

## 4.2 Occurrence rate model

The temporal occurrence of seismic events is assumed to follow a truncated Gutenberg-Richter (GR) model:

$$N = 10^a * (10^{-b*m} - 10^{-b*Mmax}) \tag{1}$$

Where $N$ is the annual rate of earthquakes with magnitude greater than $m$, $a$ is the rate of earthquakes with magnitude greater than 0 (or the productivity), $b$ is a parameter quantifying the relative distribution of small-to-large magnitude earthquakes, and *Mmax* is the maximum expected magnitude. A lower magnitude cutoff (*Mmin*) is introduced solely when applying this

relationship in the hazard integral, which will be discussed later. Since the relationship is cumulative with respect to increasing
magnitudes, this lower truncation does not affect the formulation in Equation (1) or its calibration.

Under this assumption, the GR parameters (a and b values) were estimated for each tectonic group and source zone by fitting observed annual rates from the declustered earthquake catalogue to incremental (non-cumulative) magnitude bins using a linear least-squares (LLS) method. For declustering, well-established window-based approaches were employed, including those proposed by Gardner and Knopoff (1974), Uhrhammer (1986), and Grunthal (1985). These methods were selected due

to their suitability for the regional data set and their widespread use in similar studies, as detailed in Poggi et al. (2024).

Calibration followed a two-step procedure. First, a separate occurrence model was characterized for each of the major source groups to determine regional b-values. Then, the productivity (a-value) of each zone was characterized individually by prescribing the (fixed) b-value of the corresponding group. The strategy of setting a regional b-value for large zones in advance is quite common and has been widely utilised in both research studies and industrial applications. This approach is typically

necessary when the recorded seismicity is insufficient to allow for a more detailed evaluation. Several pertinent published examples include studies by Vilanova and Fonseca (2007), Ullah et al. (2015), Ghasemi et al. (2020), and Ghione et al. (2021). Additionally, the same methodology was previously used in various regions of the current GEM Global Earthquake Hazard Model (Pagani et al., 2020; Johnson et al., 2023). Notable applications include the East African Rift (SSA, Poggi et al. 2017), North Africa (NAF, Poggi et al., 2020), and Russia/Mongolia (NEA). Because calibration of the b-value is generally

problematic, especially for zones with limited extent and a limited number of earthquake events, such a two-step procedure proved particularly useful for stabilizing results and thus obtaining more reliable productivity estimates.

The observed annual event rates were obtained from the catalogue by defining in advance the completeness periods for the different magnitude ranges. Completeness analysis was performed manually for each source group by iteratively modifying the completeness matrix and comparing the quality of the fit from GR until a satisfactory solution was obtained (see Sect. S1

for a summary of the completeness matrix defined for each group). It is important to note that the general validity of the GR relationship is often assumed to extend also to magnitudes below the completeness magnitude, which merely defines the data range used for calibrating the relationship's coefficients, without restricting the overall applicability of the formulation. It should be additionally noted that with LLS fitting the width of the non-cumulative magnitude bins is not required to be uniform, allowing greater flexibility in defining the completeness periods in the different magnitude ranges. In fact, the LLS method

remains effective with non-uniform binning because it optimises the fit by minimising the sum of the squared residuals between the observed and predicted individual rate values, independent of the calibration interval. In general, it is convenient to use uneven binning for low-seismicity regions or catalogues of limited temporal extent, where the intervals become progressively wider with increasing magnitude, for example, according to a logarithmic scheme. This approach ensures a comparable amount of calibration data for the calculation of rates, particularly for the longer return periods.

The lower magnitude truncation (Mmin) of the GR relation was set at 4.5 for all sources, a value generally accepted as the lowest intensity capable of causing significant damage to standard structures. The reason for introducing a lower truncation in the rate models is to avoid unnecessary integration steps in the hazard integral. While severe damage has occasionally been

reported from earthquakes with a magnitude of less than 4, these cases are typically associated with high frequency accelerations due to site conditions and highly vulnerable buildings. In general, such events cause only light to moderate and non-structural damage. For damage levels D4-D5 (severe damage up to collapse) of the European Macroseismic Scale (EMS-98), a magnitude range of 4.0-4.5 is generally considered appropriate, taking into account average exposure and rock conditions. Setting this threshold prevents calculations that do not significantly influence the outcome of the impact (e.g., Bommer and Crowley, 2017; Azarbakht, 2024). Furthermore, magnitudes below 4.0 do not usually contribute significantly to the hazard controlling scenario for the exceedance probabilities commonly used in engineering practise (e.g. 10% in 50 years).

Complementary to this, the upper truncation (Mmax) is defined as the largest earthquake potentially generated from the source. The definition of an optimal Mmax remains a topic of ongoing debate. The direct use of geological constraints must be carefully considered. For studies focusing on specific known structures, considering the maximum extent of the rupture is a reasonable approach (e.g., Mignan et al., 2015). However, in the present case, the mapped seismogenic structures are generally not constrained with such detail. Individual fault lines could represent one or more complex systems, and information on the actual segmentation is generally lacking. This lack of detail could lead to a significant overestimation of the expected maximum magnitude. For example, when using the AFEAD dataset (Bachmanov et al., 2017), many mapped faults may yield unrealistic magnitudes if scaling relationships are applied to their entire extent to convert the rupture area to a moment magnitude.

Although statistical algorithms exist for objectively estimating Mmax (e.g., Kijko, 2004; Kijko and Singh, 2011), their performance can be questionable under certain conditions. In fact, some instability could arise from several factors. Firstly, the rarity of large earthquakes means that statistical estimates of Mmax are highly sensitive to the inclusion or exclusion of a few extreme events. Secondly, the inherent uncertainties in the seismic catalogues, such as incomplete historical records and varying detection thresholds over time, can lead to significant variability in Mmax estimates. Lastly, these algorithms often rely on assumptions about the underlying statistical distribution of earthquake magnitudes that may not hold true in all tectonic settings, leading to further instability and potential overestimation or underestimation of Mmax.

For these reasons, we have chosen a simpler, but rather conservative and at the same time defensible approach. In practice, Mmax was set as the maximum observed magnitude plus an increment of 0.4 units. The value of the increment was chosen as the highest value that still provided physically credible earthquakes for the entire region, also taking into account the standard uncertainty in magnitude estimation, especially for historical events. An additional deviation of ±0.1 units was then allowed in the hazard calculation (see Logic Tree section) to account for the epistemic uncertainty associated with the definition of the magnitude increment. It should be noted that the correct definition of Mmax is particularly critical for ground motion levels with very low exceedance probabilities (i.e., fairly long return periods), which are generally relevant for special structures and critical facilities. For these, a more critical review of the operational definition of Mmax may be required.

A summary of the derived G-R seismicity parameters calibrated for each tectonic source group is shown in **Figure 4**.

## Group A

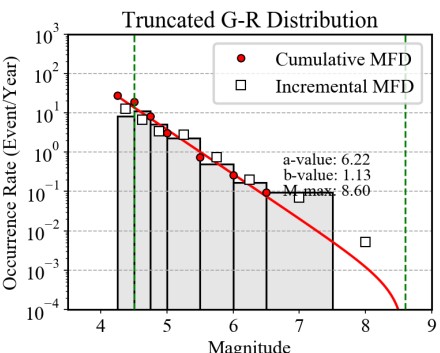

## Group B

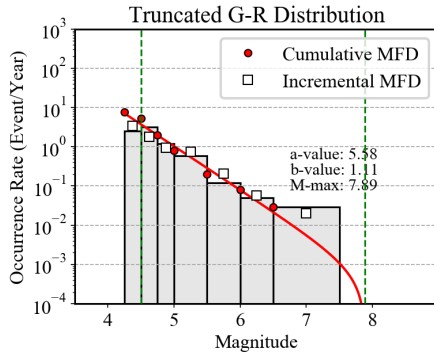

## Group C

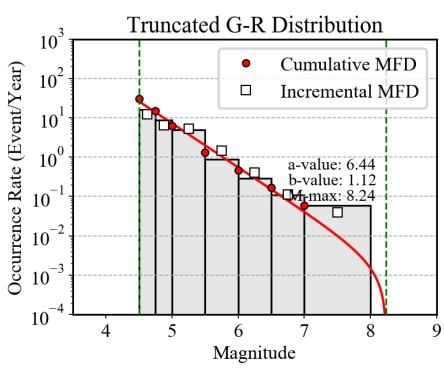

## Group D

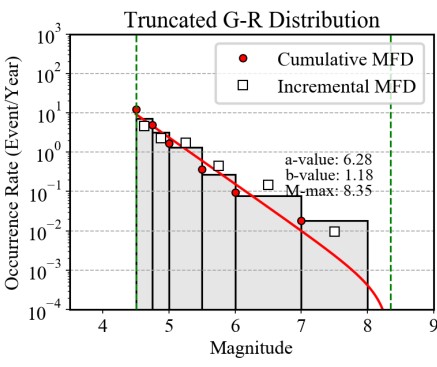

## Group E

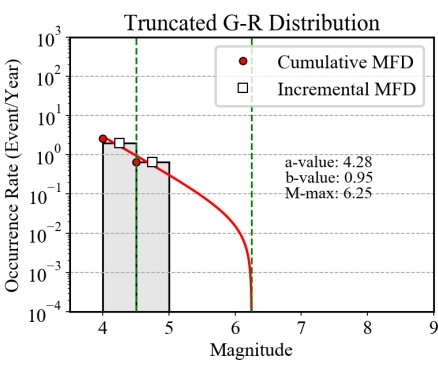

## Group F

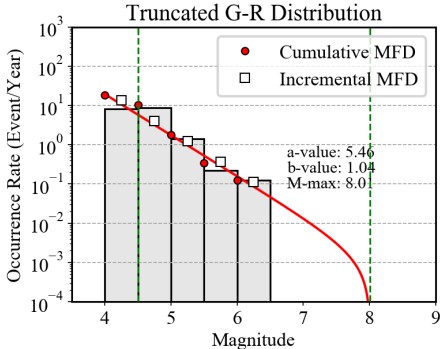

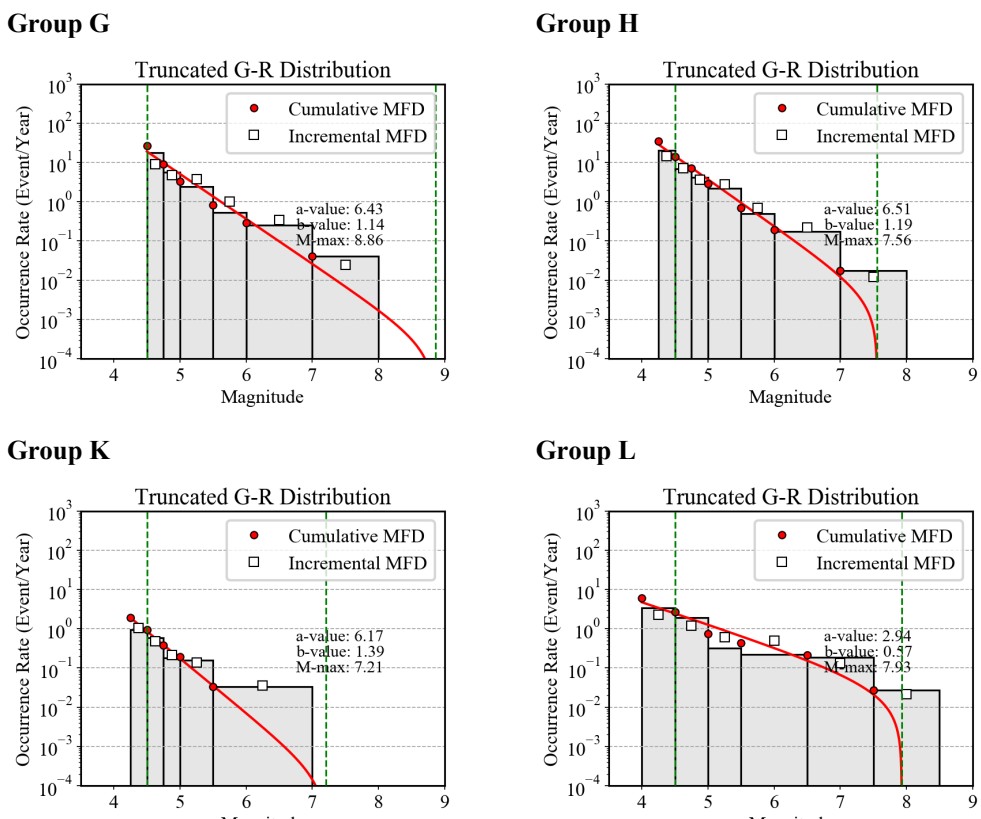

**Figure 4.** Gutenberg-Richter occurrence relations calibrated for the different source groups of the Central Asia model. White squares and red dots are the observed incremental and cumulative occurrence rates, respectively, while the grey histogram and the red line represent the incremental and cumulative rates from the inverted Gutenberg–Richter relation. The minimum and maximum truncation magnitudes are indicated as grey dashed vertical bounds. The width of the incremental bins corresponds to that defined in the completeness matrices of Sect. S1.

## 4.3 Rupture mechanism

A key feature of OpenQuake is the ability to model individual earthquake events as ruptures of finite extent by simulating the spatial orientation and kinematics of each fault given a specified fracture mechanism. This is very advantageous when using modern generation ground motion prediction models that are capable of using fault-dependent distance metrics (e.g., Rjb, Rrup, see Douglas 2003 for a comprehensive discussion) and mechanism-dependent calibration coefficients. However, the major drawback is that the probability distributions of rupture mechanisms must be defined for each source (or group of sources), which is only possible if sufficient seismotectonic information is available for the region.

To define the predominant rupture mechanism of each source zone of the Central Asian model, we combine the available information from mapped surface faults (see Poggi et al., 2024), especially for the strike direction, with moment tensor

solutions from the GCMT bulletin (Ekström et al. , 2012). For the study region, 814 focal mechanisms are available for events in the range of 4.64 < Mw < 7.61. Geometric parameters (strike and dip) of the different source zones were characterized by analysing the geometry of the focal mechanism using the "beachball" representation (see **Figure 5**), while the dominant fault style was accessed by examining the distribution of B-T axis orientations using the classification diagrams of Kaverina et al.

(1996) (**Figure 6**), as implemented in the FMC code of Álvarez-Gómez (2019).

For the implementation of rupture mechanisms in OpenQuake, the predominant faulting style is represented by a combination of dip and rake angles (**Table 1**), following the formalism described by Aki and Richards (1980). When multiple faulting styles are identified, weights are assigned according to the relative proportions of clusters in the Kaverina's diagram. For example, in Group D, there are two main mechanisms: reverse and strike-slip, with a similar number of reported solutions. This leads to

an initial probability fractionation of about 50-50%. However, defining the actual strike direction from the mapped faults was ambiguous, resulting in the identification of two main families of orientations. Since there was no evidence suggesting the dominance of one family over the other, we further split the original 50% probability into 25% for each orientation. Similar considerations were made for the other groups. From the analysis, as expected, it appears that throughout the area there is a majority of reverse style mechanisms, with a small, though not negligible, contribution from strike-slip events. A proportion

of normal style mechanisms is also evident (e.g., Group C and F), but less significant.

A summary of the rupture mechanisms associated with each zone group is given in **Table 2**.

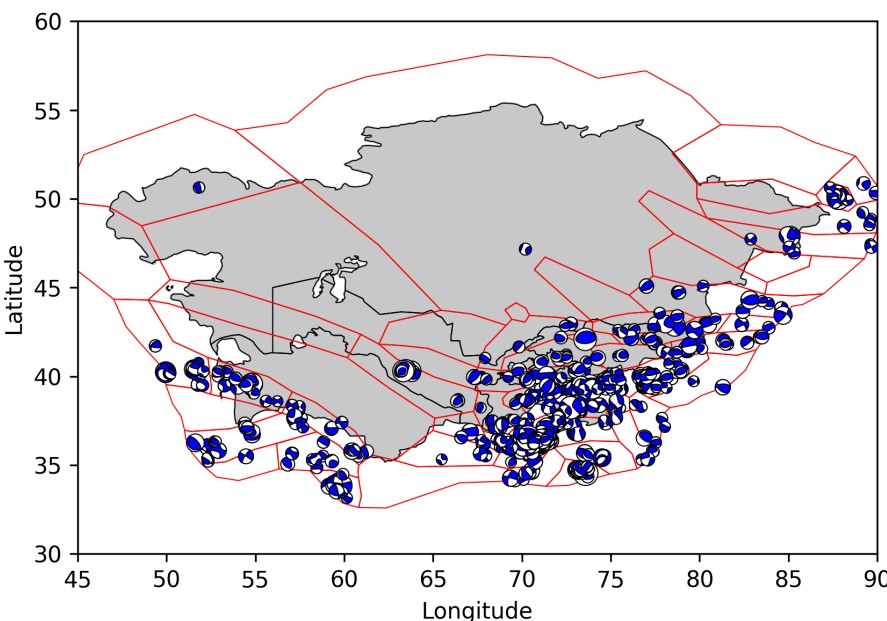

**Figure 5.** Distribution of "beachballs" of the 814 events available from the GCMT catalogue for the region. Traction axis is conventionally represented in blue. Plot was created using the Obspy Python library (Beyreuther et al., 2010, https://github.com/obspy).

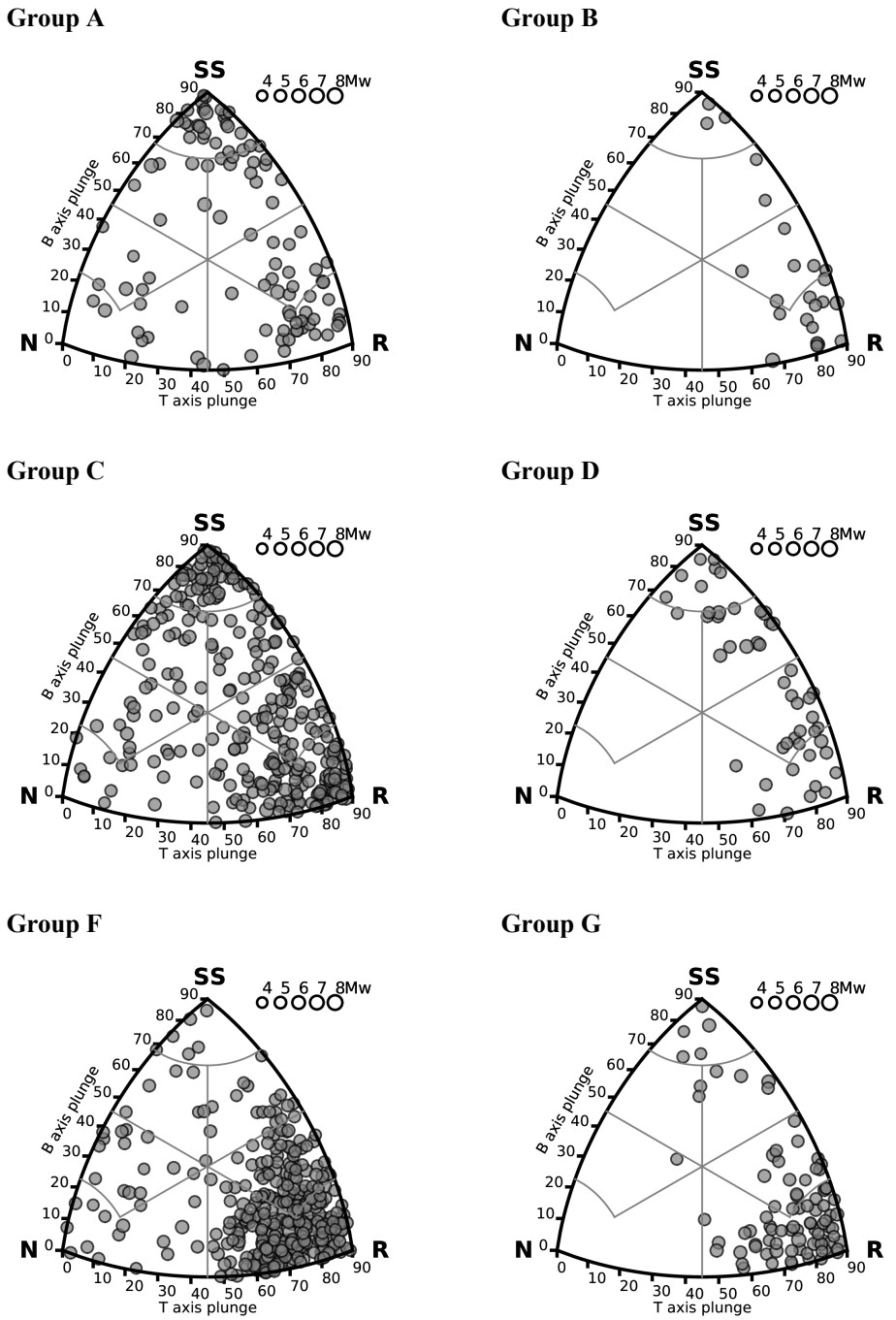

**Figure 6.** B-T axis classification of the GCMT moment tensor solutions available for each source group of the shallow seismicity model (due to lack of events, group E is not shown).

| Fault style | Standard dip (deg) | Standard rake (deg) |
|---|---|---|
| **Reverse** | 45° | 90° |
| **Normal** | 60° | -90° |
| **Left-lateral strike slip** | 90° | 0° |
| **Right-lateral strike slip** | 90° | 180° |

**Table 1**. Conversion table between general faulting style and the geometrical fault parameters dip and rake as used in OpenQuake.


| Grou | Probability | Strike | Dip | Rake |
|---|---|---|---|---|
| **A** | 0.4 | 60° | 45° | 90° |
| | 0.2 | 120° | 45° | 90° |
| | 0.4 | 120° | 90° | 180° |
| **B** | 0.6 | 120° | 45° | 90° |
| | 0.4 | 120° | 90° | 180° |
| **C** | 0.5 | 70° | 45° | 90° |
| | 0.4 | 120° | 90° | 180° |
| | 0.1 | 30° | 60° | -90° |
| **D** | 0.25 | 70° | 45° | 90° |
| | 0.25 | 120° | 45° | 90° |
| | 0.5 | 120° | 90° | 180° |
| **E** | 0.5 | 70° | 45° | 90° |
| | 0.5 | 120° | 90° | 180° |
| **F** | 0.7 | 70° | 45° | 90° |
| | 0.3 | 30° | 60° | -90° |
| **G** | 0.8 | 80° | 45° | 90° |
| | 0.2 | 120° | 90° | 180° |
| **H** | 1.0 | 70° | 45° | 90° |
| **K** | 1.0 | 120° | 45° | 90° |
| **L** | 1.0 | 70° | 45° | 90° |

**Table 2.** Summary of the rupture mechanisms assigned to each tectonic group with relative probability.


## 4.4 Additional model parameters

The source zones and calibrated seismicity parameters were used to create the homogenous areas source model in xml format
using the Python utilities available from the Hazardlib library of OpenQuake. Additional parameters needed for the calculation
were provided, such as:

- the magnitude scaling relation (Wells and Coppersmith, 1994), which used to numerically constrain the subsurface
  length (L) and width (W) of the earthquake ruptures;
- the fault rupture aspect ratio (1:2);
- the upper and lower seismogenic depths required to constrain the extent of the rupture surfaces in each hypocentral
  domain (see **Table 3**);
- the distance interval used to discretize the area source model into a finite grid of sources (10 km spacing).

| Depth layer | Lower seismogenic depth (LSD) | Upper seismogenic depth (USD) |
|---|---|---|
| **Shallow depth sources** | 0km | 65km |
| **Intermediate depth sources** | 35km | 200km |
| **Deep sources** | 150km | 350km |

**Table 3.** Lower and upper seismogenic depths adopted to constrain the rupture extension in the different source depth layers.

It must be noted that the values in **Table 3** are strictly derived from the depth limits defined for the source zones, based on the
seismicity analysis performed in Section 4.1. To define the Lower Seismogenic Depth (LSD) and Upper Seismogenic Depth
(USD) boundaries, we have practically allowed ruptures occurring at the interface between different depth zones to extend to
a certain limit, which ranges between 15 and 30 km, depending on the expected magnitudes. It is important to recognize that
LSD and USD are not exact values, but rather conservative limits intended to prevent the development of ruptures with
unrealistic depth extents.

## 5 Smoothed seismicity model

When calculating earthquake hazard using homogeneous source zones, it is assumed that the probability of occurrence is
spatially the same within areas. This assumption is particularly advantageous in regions with short and/or incomplete
earthquake records because it accounts for earthquakes that occur at potential locations not yet represented in the catalogue.
However, the approach may not be suitable for regions where seismicity is known to be well localized along major tectonic

structures and specific domains. This is the case in the study region, particularly along the southern active collisional margin,

where analysis of the earthquake catalogue confirms the presence of nonuniform spatial patterns of seismicity that are closely related to the development of specific seismotectonic features (e.g., thrusts). The associated smearing effect could, for example, lead to underestimation of the calculated hazard at some locations near the localized seismicity and overestimation at other, more distant locations. To overcome this limitation, the smoothed seismicity approach has been introduced (e.g., Frankel, 1995), in which the calculated event rates are spatially reorganized to follow the observed earthquake pattern.

In this study, we use a variant of the smoothing procedure proposed by Poggi et al. (2020), which has the great advantage of preserving the overall equilibrium of rates in each individual zone. The degree of smearing of the rates is controlled by the smoothing length parameter ($\lambda$), which reflects the belief in the actual observed seismicity pattern. The larger $\lambda$ is, the more uniform the pattern of event rates will be, ideally converging to uniform zonation. Conversely, a small value of $\lambda$ will accurately reflect the observed seismicity pattern.

However, determining an optimal smoothing length is difficult and requires some expertise. Because $\lambda$ is a parameter subject to few constraints in the model and therefore contributes to its epistemic variability, several alternative values (one central value and two edge cases) were used in a logic-tree approach with assigned triangular weights. In addition, to avoid typical "bull-eye" smearing effects in zones with too few observed events (e.g., in the Kazakhstan cratonic shield), a different combination of smoothing lengths was used for regions with high and low seismic productivity. High $\lambda$-values were also used

for the deep seismicity zones where the uncertainty about location is large. See **Table 4** for the combination of smoothing lengths for each group. The smoothing procedure was applied separately to the shallow, intermediate, and deep seismicity layers (see, for example, **Figure 7**).

|  | Smoothing length ($\lambda$) | Weight | Apply to region |
|---|---|---|---|
| Low seismicity zones + Deep sources | 25 | 0.25 | B, E, H, K, L |
|  | 50 | 0.50 |  |
|  | 100 | 0.25 |  |
| High seismicity zones | 10 | 0.25 | A, C, D, F, G |
|  | 20 | 0.50 |  |
|  | 40 | 0.25 |  |

**Table 4.** Combination of smoothing length ($\lambda$) parameter adopted for regions of low and high seismicity of the Central Asia model, and associated weights.

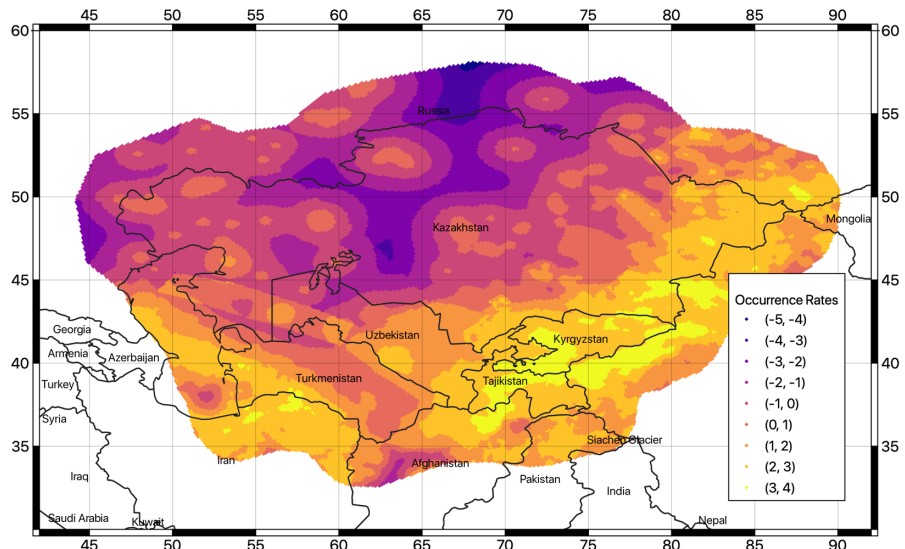

**Figure 7.** Spatially variable occurrence rates using the smoothing approach for the source layer at shallow-depth. Rates shown are from a weighted average of the three smoothing length values in **Table 4**. Units are expressed as the logarithm of the annual occurrence rate (per grid cell) of events greater than zero, to highlight the differences in visualization.

## 6 Finite fault model

The use of standard distributed seismicity models has the advantage that a wide range of possible earthquake scenarios can be included in the calculation. Nonetheless, peculiarities of specific sources may be lost, which is particularly inconvenient when the near-field ground motion level is target. An alternative to partially overcome this limitation is to include finite (3d) fault sources in the source model (e.g., Danciu et al. 2018). Starting from a homogenous regional dataset of potentially active faults (see Poggi et al., 2024 for a detailed description of the input datasets used) that includes information from geologic studies, scientific literature and local databases, the fault source model is then built assuming an occurrence model and appropriate seismicity parameters (e.g., scaling relationships, aseismic coefficient and seismogenic depths) using an ad-hoc Python fault modelling tool developed as part of the Model Building Toolkit from GEM (https://github.com/GEMScienceTools/oq-mbtk). In the absence of clear evidence of "characteristic" model behaviour, we use a simple double-truncated Gutenberg-Richter distribution to model earthquake occurrence on faults, consistent with the event model assumed for distributed seismicity. Following the assumptions originally formulated by Schwartz and Coppersmith (1986), occurrence rates (as a-values) of each fault are here derived directly from the slip rate estimates by balancing the scalar seismic moment accumulation rate and the scalar moment release rate (Molnar, 1979) from the integral of the incremental MFD while using a direct fitting procedure (Poggi et al. 2017). Here, we assume a default shear modulus of about 30 GPa (e.g., Bird and Liu, 2007) and an aseismic coefficient of 0.1 to account for the accumulated seismic moment released aseismically by creep and plastic deformation.

The calibration of this last parameter is challenging due to the lack of consensus within the seismological and geodetic communities on its optimal value, and the absence of direct methods for its evaluation. Moreover, the literature on this topic is quite limited. Setting the aseismic coefficient too close to 0 would be unrealistic, as a portion of the slip is inevitably released through plastic deformation. Conversely, values greater than 0.2 often result in inconsistencies between the total moment derived from slip rates and that observed from seismic events in the catalogue. Through comparative analyses of hazard levels derived from fault models and seismic catalogues, we evaluated different values for the aseismic coefficient. A value around 0.1 was found to be practical, balancing the need for realism with the constraints imposed by the available data and modelling techniques. Although this value is an approximation, it aligns reasonably well with empirical observations and ensures consistency between the modelled hazard levels and the observed seismic activity.

The b-value and maximum magnitude generated are derived a priori from the seismicity analysis of the source zone enclosing the fault. However, if the fault has a limited extent, the maximum magnitude is scaled appropriately by applying the scaling relationship of Leonard (2014) to avoid unrealistically large magnitudes.

The derived fault source model currently contains 1444 individual fault segments (**Figure 8**), covering most of the active shallow crust currently affected by seismicity. However, it must be emphasized that the fault source model alone may not be complete enough to fully represent all shallow seismicity, especially at low magnitudes and large depths, and therefore cannot be used as an alternative to the distributed seismicity model. To fill in possible missing events, background source layers were added to the fault model during computation. The background model was carried over from the homogeneous zonation model (for shallow, intermediate, and deep sources), but the maximum magnitude generated for the shallow zones was limited to 6. Ruptures above this threshold are considered to show a clear surface expression and therefore should be adequately represented in the fault database. Intermediate and deep sources remain unchanged.

It should be noted that in a source model calibrated using slip rates, there is no general guarantee that the overall earthquake rate balance will match with that calculated from observed seismicity, although some degree of agreement would of course be desirable. Indeed, there are practical problems that limit direct comparison. These include the definition of the extent of the area around the fault used to integrate rates from the distributed source model. As we have tested, appropriate tuning of the area of integration would result in an artificially induced good match, rendering the direct comparison of rates useless. Direct assignment of earthquakes to fault lines is also nontrivial. Therefore, the most appropriate verification strategy is to compare the final hazard results of both the fault and distributed seismicity models. However, even in this case, perfect agreement is not the goal, since the models are likely to produce complementary results, but a general agreement in hazard levels is expected.

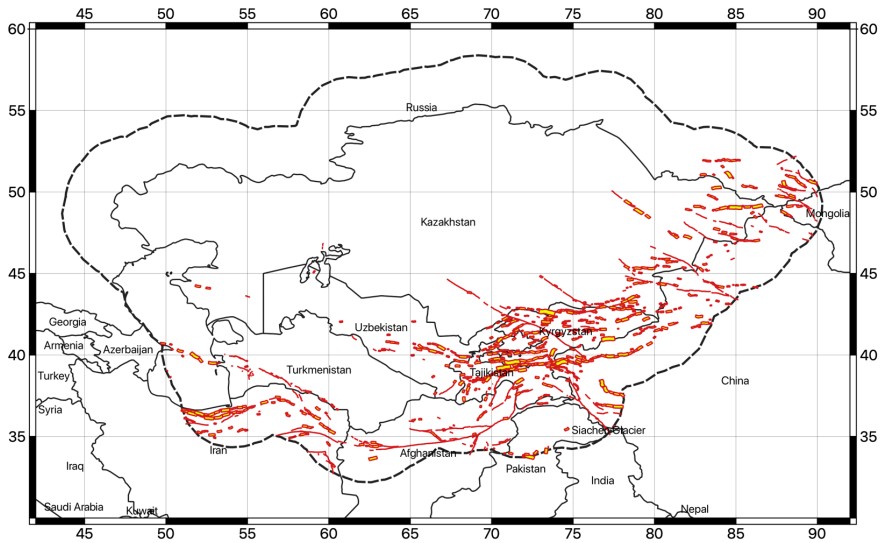

**Figure 8.** 3D geometry of the faults in the final source model. Surface fault traces are shown in red, while the surface projection of the fault plane is in yellow.

## 7 Ground motion model

Calibration of the ground motion prediction model is an important aspect of a hazard model development. Although few studies

have been conducted for the area, there is a general lack of locally calibrated models that can be used to predict the complete set of target response spectral accelerations. To overcome this limitation, a set of external ground motion prediction equations (GMPEs) must be used. Preferably, the most appropriate GMPEs should be selected by direct comparison with local earthquake recordings in a magnitude and distance range that is meaningful for the analysis. However, if no or too few empirical earthquake observations are available, indirect selection criteria should be used, as described in Cotton et al. (2006). The

criteria include:

- analysis of the performance of the ground motion model
- characteristics of the calibration dataset (type, quality, and coverage range of the data).
- compatibility of target tectonic setting with that of the model
- suitability of the functional form (availability of the information required for the predictor variables, consistency of

the output with respect to hazard assessment requirements).

In this work, we followed these criteria for selecting a set of appropriate ground motion models.

## 7.1 Tectonic regionalization

To account for the variability of tectonic environments in the region that is responsible for the differential attenuation of ground motions from source to site, a strategy for regionalizing ground motion modelling was employed. The first step was to identify subregions of supposedly homogeneous attenuation behaviour. For this purpose, we rely on the classification proposed by Chen et al. (2018), which combines the analysis of seismological (seismic moment rates, attenuation of 1Hz Lg coda), geological (plate boundary models, digital geological mapping), and geophysical (crustal Vs velocities) data from global datasets.

According to this classification, three seismotectonic domains are represented in Central Asia: active shallow crust, non-cratonic active stable crust and cratonic stable continental crust (**Figure 9**). On this basis and with some adjustments based on local considerations, the different zones of the shallow seismicity source models were classified accordingly into three main tectonic region types (TRT, see **Figure 10**):

- TRT 1 – Standard active shallow crust
- TRT 2 – Active stable crust
- TRT 3 – Cratonic crust

An additional fourth region (TRT4) was then added to represent the intermediate to large depth source zones.

It should be noted that the grouping used in the tectonic regionalisation generally differs from the zonation of the source model. The grouping of the source zone was based on similarities in the process of earthquake generation at the source level, while the grouping for the tectonic regionalisation type (TRT) was based on expected differences in general attenuation behaviour along the path from the source to the site, with the aim of differentiating the ground motion prediction models. Although there are some similarities between the two classification schemes, they are not equivalent as they are based on different assumptions and constraints.

The final step is then to select one or more ground motion prediction models for each TRT.

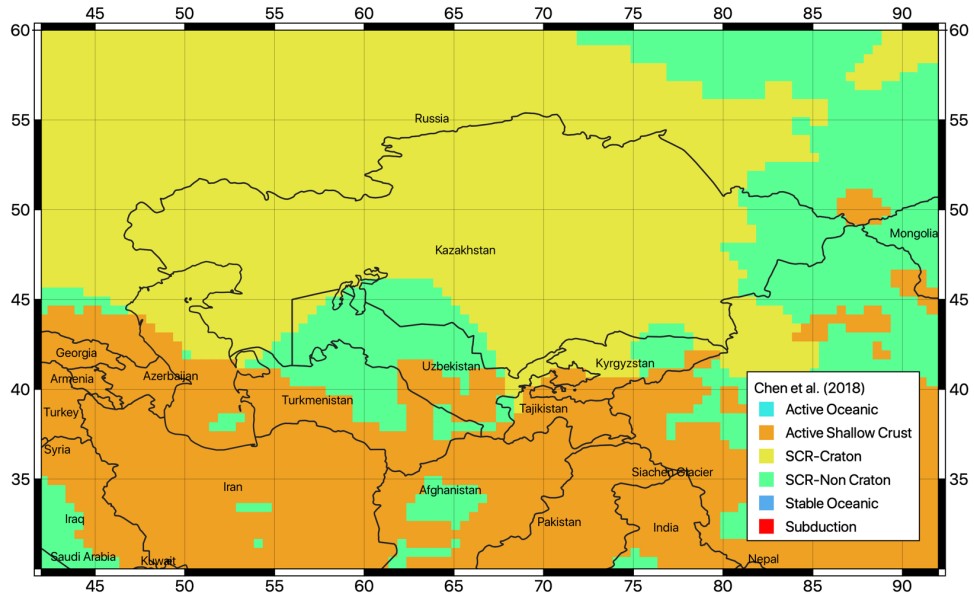

**Figure 9.** Tectonic classification proposed by Chen et al. (2018) used to guide the regionalization of the ground motion prediction model for Central Asia.


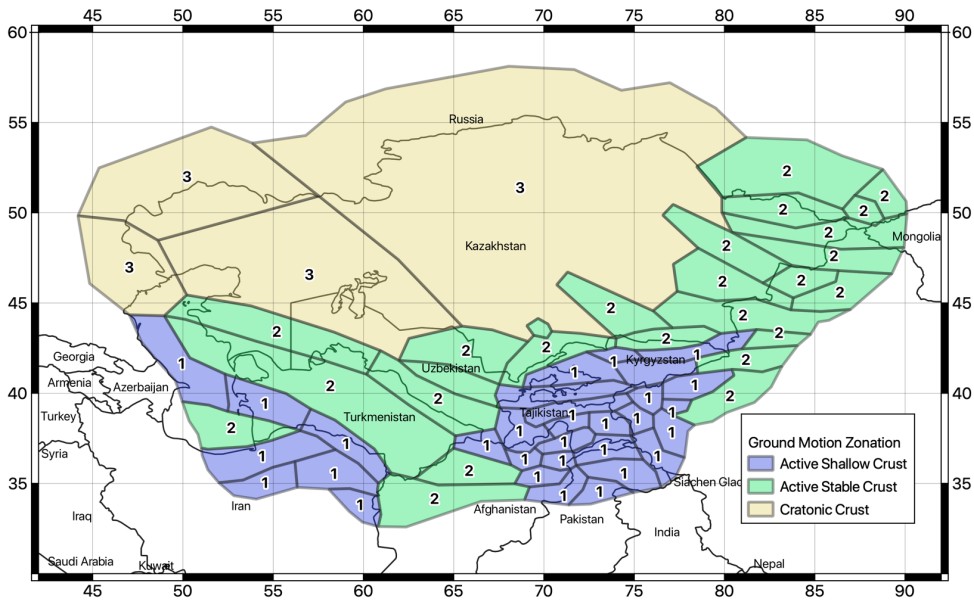

**Figure 10.** Tectonic region type (TRT) classification of the source zones of the Central Asia model.

## 7.2 GMPE selection

In a first step, ground motion models compatible with the identified TRT were isolated from the ground motion model library of OpenQuake (the HazardLib). Following the selection criteria recommended by Cotton et al. (2006) and the studies recommended by the local experts of the consortium, the number of suitable models was limited to the five most representative for the study region. Two equally weighted ground motion models were selected for the shallow tectonic conditions (Active Shallow - AS and Stable Continental - SC), while only one suitable GMPE was identified for Deep Seismicity (DS). The

performance of each ground motion model was analysed for a combination of magnitudes and distances, and for the different intensity measure types required for the study (see the Trellis plots in **Figure 11**).

As for the case of model selection, ranking of the ground motion models can be performed based on their degree of agreement with observed calibration data, such as strong motion recordings. Efficient ranking methods, such as the LLH approach proposed by Scherbaum et al. (2009) or the EDR by Kale and Akkar (2013), are commonly used for this purpose. However,

in our case, the available records for the region were insufficient to perform a robust data-driven evaluation. Therefore, a more conservative approach to assigning weights was necessary.

Assuming that active shallow crust (TRT1) can be solely represented by AS models and cratonic crust (TRT3) by SC models, we chose to represent stable crust conditions (TRT2) as an intermediate combination of the AS and SC models. This operational choice is justified by the assumption that purely cratonic attenuation behaviour is unlikely in regions such as the Kazakh Shield,

due to its significant tectonic reworking, interactions with active tectonic boundaries and complex crustal composition (e.g., Molnar & Tapponnier, 1975). Given that TRT2 should exhibit less extreme attenuation behaviour compared to standard regions with active shallow crust, it can be inferred that buffer tectonic regions surrounding large active structural systems, such as the Tianshan Massif and Turkmenistan, would exhibit intermediate behaviour. Nonetheless, this operational choice can only be fully verified if sufficient ground motion records are available for comparison with the selected ground motion models.

The main advantage of such a two-step weighting procedure (for ground motion models and tectonic groups) is that it leads to smooth and regionally variable ground motion predictions, thus avoiding sharp variations between adjacent tectonic environments. The hybrid region allows the blending of ground motion models, effectively accounting for intermediate tectonic conditions. While the same goal could have been achieved by directly assigning a weight of 0.25 to each of the four models in a one-level zonation, a two-level zonation provides a level of abstraction to better deal with mixed regions.

Furthermore, additional and/or different ground motion models or intermediate weighting (e.g., between AS and DS in TRT4) can be easily integrated by maintaining the developed logic of tectonic regionalization, facilitating future changes as new data becomes available or based on improved understanding of the area and progress in the field.

The selected GMPEs and their corresponding relative weights are then summarized in **Table 5**, while the relative weighting scheme for each tectonic group is presented in **Table 6**.


| Tectonic Id | Ground Motion Model | Weight |
|---|---|---|
| AS | Campbell and Bozorgnia (2014) | 0.5 |
| | Chiou and Youngs (2014) | 0.5 |
| SC | Pezeshk et Al. (2011) | 0.5 |
| | Atkinson and Boore (2006) – Modified 2011 | 0.5 |
| DS | Parker et Al. (2020) – for subduction interface | 1 |

**Table 5.** Selected ground motion prediction models grouped by tectonic region applicability.


| | AS | SC | DS |
|---|---|---|---|
| TRT 1 | 1 | 0 | 0 |
| TRT 2 | 0.5 | 0.5 | 0 |
| TRT 3 | 0 | 1 | 0 |
| TRT 4 | 0 | 0 | 1 |

**Table 6.** Weight combination of the GMPE groups (**Table 5**) with respect to tectonic zonation of the Central Asia model.


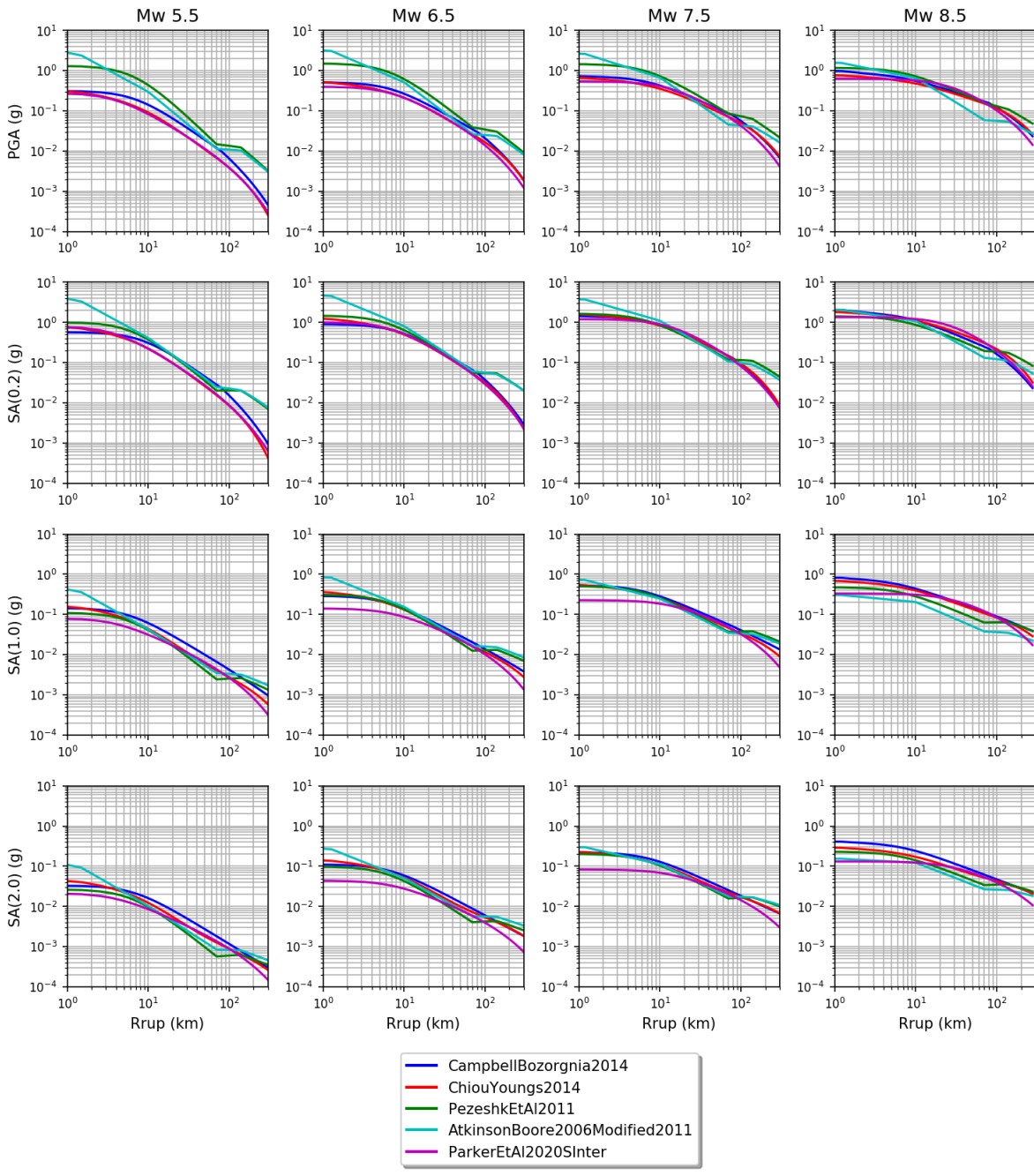

**Figure 11.** Comparison of ground motion distance attenuation for the selected prediction models for different magnitudes (columns) and intensity measure types (rows). The typical deflection of the ground motion due to refraction at the Moho interface is clearly visible in the SC crust models at about 100km.

## 8 Epistemic uncertainty and logic-tree

To account for epistemic variability in key model parameters, a logic-tree approach was used (**Figure 12**). From a technical point of view, the implemented logic-tree is divided between the two main components of the model: source characterization and ground motion modelling characterization. Each component includes different branching levels that represent either an independent uncertainty type (as in the case of b-value and Mmax) or the permutation of alternative models applied in different regions (as in the case of GMPE regionalization).

The source model part of the logic tree includes both the developed distributed (smooth) seismicity model and the faults+background model, as independent branches. The two models were weighted equally. The largest uncertainty associated with the fault model relates to the definition of the slip rate conversion from the rate classes. Therefore, to represent the uncertainty associated with this, three alternative occurrence models were included. The model that provides the median estimate, considered the most reliable, has the largest weight (0.6) while the other two marginal models have a smaller weight (0.2). Similarly, three independent distributed seismicity models were implemented using different smoothing lengths, which is currently a very subjective parameter. However, to reduce the complexity of the OpenQuake calculation, the alternative distributed models with different parametrizations were combined into a single weighted average occurrence rate model, using weights as indicated in the logic-tree table. Therefore, the variability in smoothing length is not directly represented by independent branches, although it is formally accounted for in the source model formulation. This simplification should be considered when examining the variability of hazard calculations (e.g., quantile hazard curves).

The logic-tree framework makes it possible to consider the confidence in different parameter estimates and their relative impact on seismic hazard assessments by assigning appropriate weights to the different branches. Epistemic uncertainty in the calibration of the occurrence rate model was accounted for by incorporating the variability of the Gutenberg-Richter (GR) b-value and maximum magnitude (Mmax). While the b-value is reasonably constrained by regional data, local variability can still occur. This local variability is therefore accounted for by introducing a range of plausible b-values into the model ($\pm0.05$ with triangular weights of 0.25-0.5-0.25). The choice of $\pm0.05$ reflects typical observed variations in b-values within similar tectonic settings, balancing the need for accuracy with the potential for regional discrepancies. Conversely, the uncertainty in Mmax is considered conservatively to compensate for the less reliable calibration of this parameter by accounting for an additional $\pm0.1$ (again with triangular weights of 0.25-0.5-0.25). The $\pm0.1$ range for Mmax was selected based on the potential variability in the historical earthquake record and the inherent uncertainties in defining the upper limits of earthquake magnitudes, as discussed in Section 4.2. This conservative approach helps to minimise the risk of underestimating the potential for larger, albeit less frequent, earthquakes, thereby enhancing the defensibility of the seismic hazard estimates.

The ground motion logic tree consists of four branching levels, each representing a particular combination of ground motion prediction model groups (SA, SC and DS) applied to the different regions (TRT, see section on ground motion regionalization). It must be emphasized that such a grouping approach, although it may seem complex at first sight, allows greater flexibility in

defining regions with intermediate attenuation behaviour, since a heterogeneous combination of different tectonic groups is
possible (see **Figure 12**).

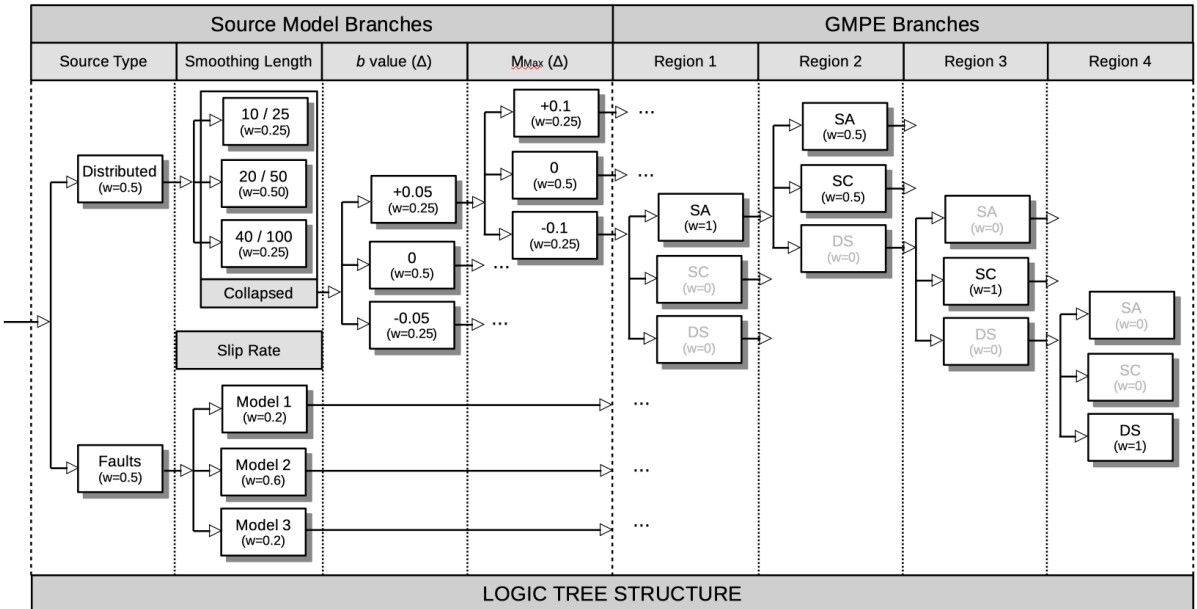

**Figure 12.** Schematic representation of the logic-tree structure of the hazard model for Central Asia, which includes four levels of
branching to account for uncertainties in both the source model and the ground motion models.

## 9 PSHA results

The investigated area consists of a mesh of 8028 sites on a regular grid with a spacing of at 0.2 degrees (about 20 km). For
each site in the grid, free rock conditions are assumed with a fixed reference value of shear-wave velocity average over 30
meters (Vs30) of 800 m/s, corresponding to class A (standard rock) in the classification of Eurocode8 (CEN 2004) and NERHP
(BSSC 2003). All calculations for this study were performed using version 3.11 of the OpenQuake engine, available at
https://github.com/gem/oq-engine/tree/engine-3.11 (last accessed on 16/08/2021).

Ground motion probability of exceedance (PoEs) for a given observation time are computed for PGA and for 5%-damped
response spectral acceleration at 0.1s, 0.2s, 0.5s, 1s, 2s, and 3s (the oscillation periods allowed by the selected ground motion
models). As is often the case, the integration of ground motion was truncated at 3 sigma of the median prediction. Calculation
results are provided in the form of a) mean and quantile (0.05, 0.15, 0.5, 0.85 and 0.95) hazard curves for each intensity
measure type (Imt) and site (see **Figure 13** and **Figure 14** for example results calculated for six selected sites), b) Uniform
Hazard Spectra (UHS, **Figure 15**) and c) hazard maps calculated for return periods of 25, 50, 100, 250, 475, 500 and 1000
years, corresponding to a probability of exceedance of 86, 63, 39, 18, 10, 9 and 5% in 50 years of observation (examples in
Sect. S2). It should be noted that shorter return periods could not be calculated when approaching 100% PoE due to numerical

limitations. Calculations were performed assuming a Poisson earthquake occurrence model. See **Figure 16** for an example map of PGA.

Comparing the uncertainty of the hazard curves and UHS, the same variability between sites can be observed. In general, a large scatter in the hazard curves is evidence of high epistemic uncertainty in the model variables as depicted in the logic tree, while a low scatter indicates that these uncertain parameters have limited sensitivity to the computed hazard, which is

conversely controlled by model components that are "assumed" to be more reliably constrained (i.e., without associated epistemic uncertainty). In Dushanbe, low dispersion is visible for PGA (and 10%PoE), but as shown in the associated UHS, larger uncertainty is associated with other spectral ordinates, with opposite trends in Bishkek, for example. Such complementary behavior is often, but not only, related to the variability of the ground motion model predictions.

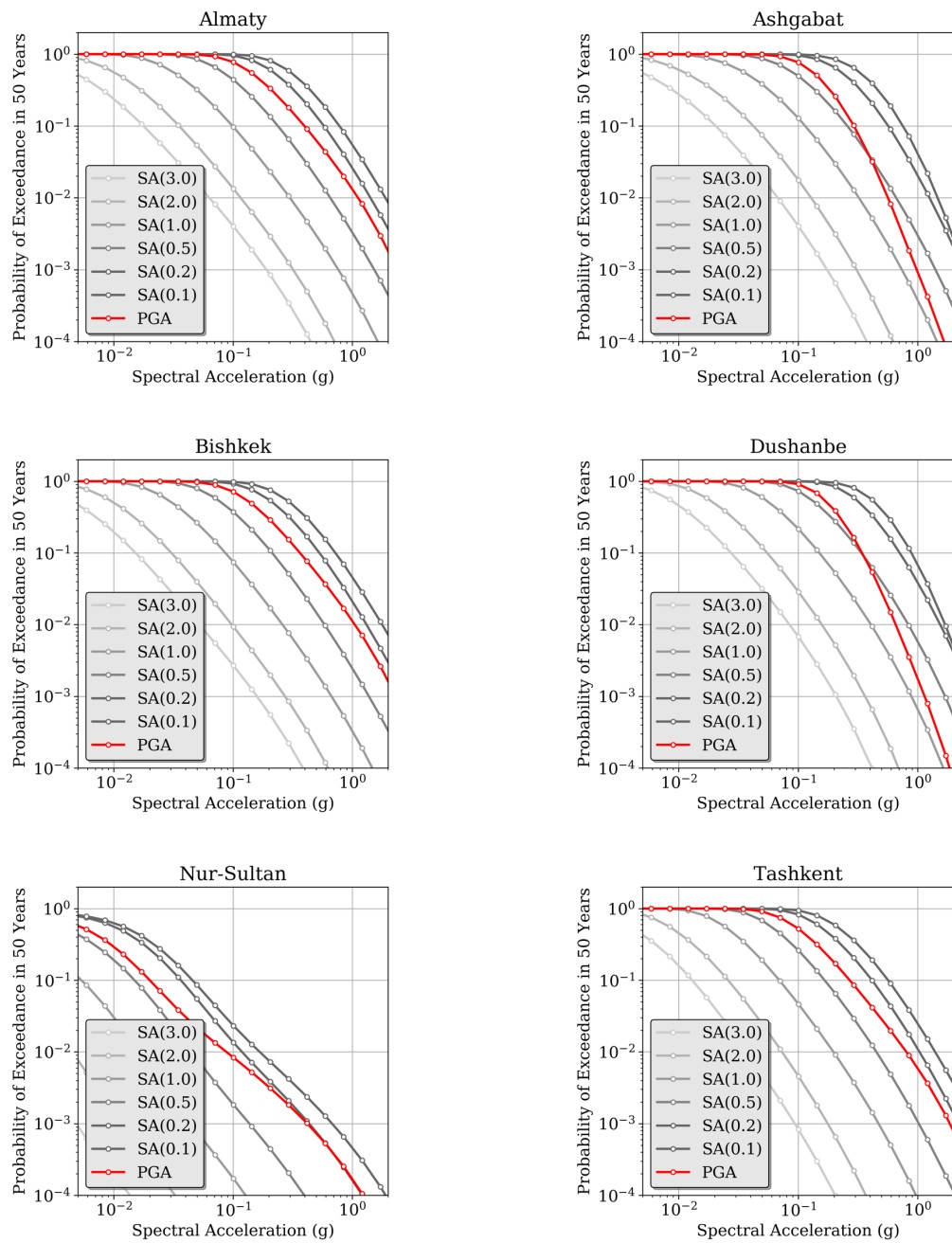

**Figure 13.** Example of mean hazard curves calculated at six selected target sites (all state capitals plus Almaty, Kazakhstan; note that Nur-Sultan was formerly known as Astana) for different intensity measure types (PGA and spectral accelerations for periods from 0.2 s to 3 s) with a 10% probability of exceedance in 50 years.


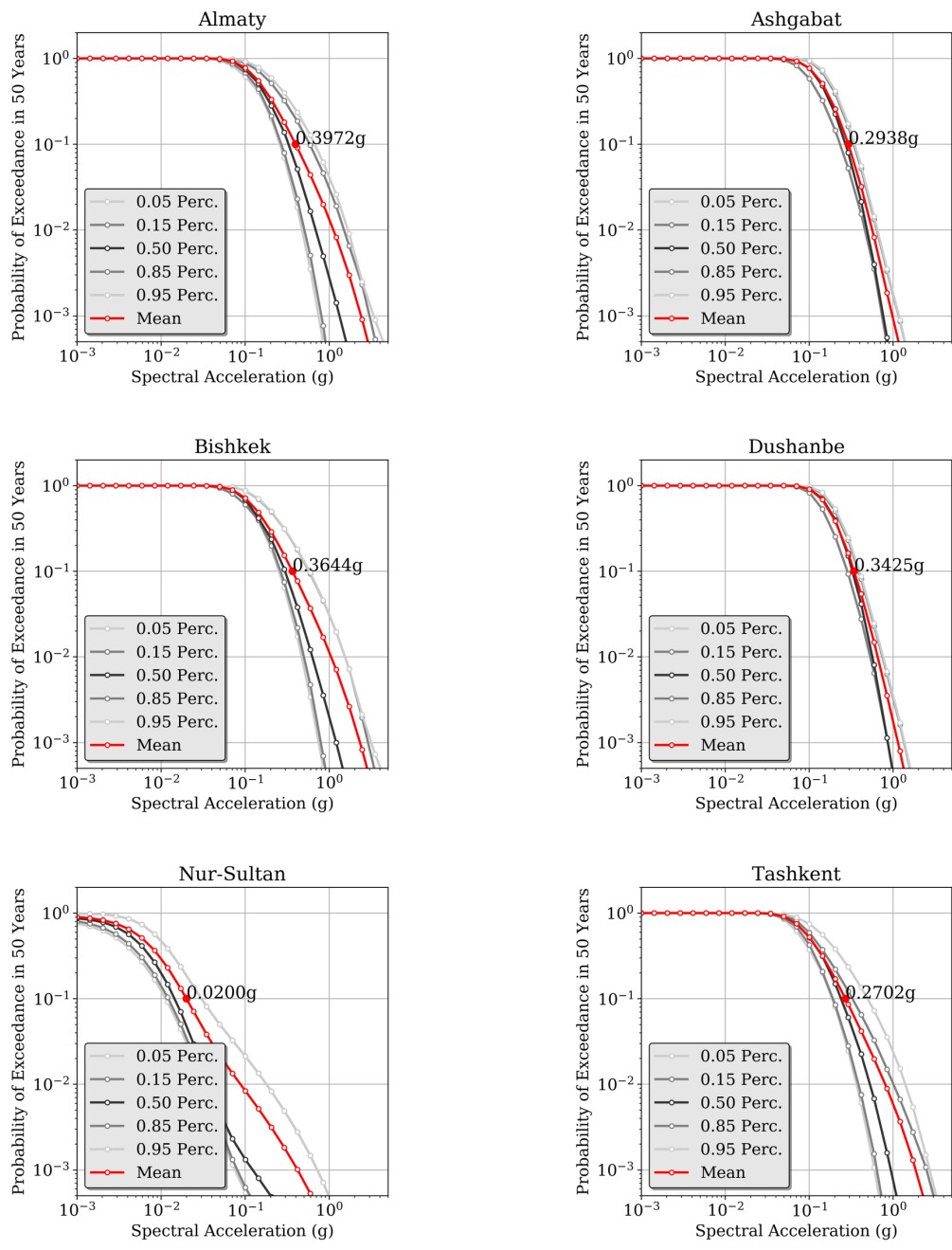

**Figure 14.** Example of mean hazard curve statistics (mean and quantiles) calculated at six selected target sites (all country capitals plus Almaty, Kazakhstan) for PGA with a 10% probability of exceedance in 50 years.

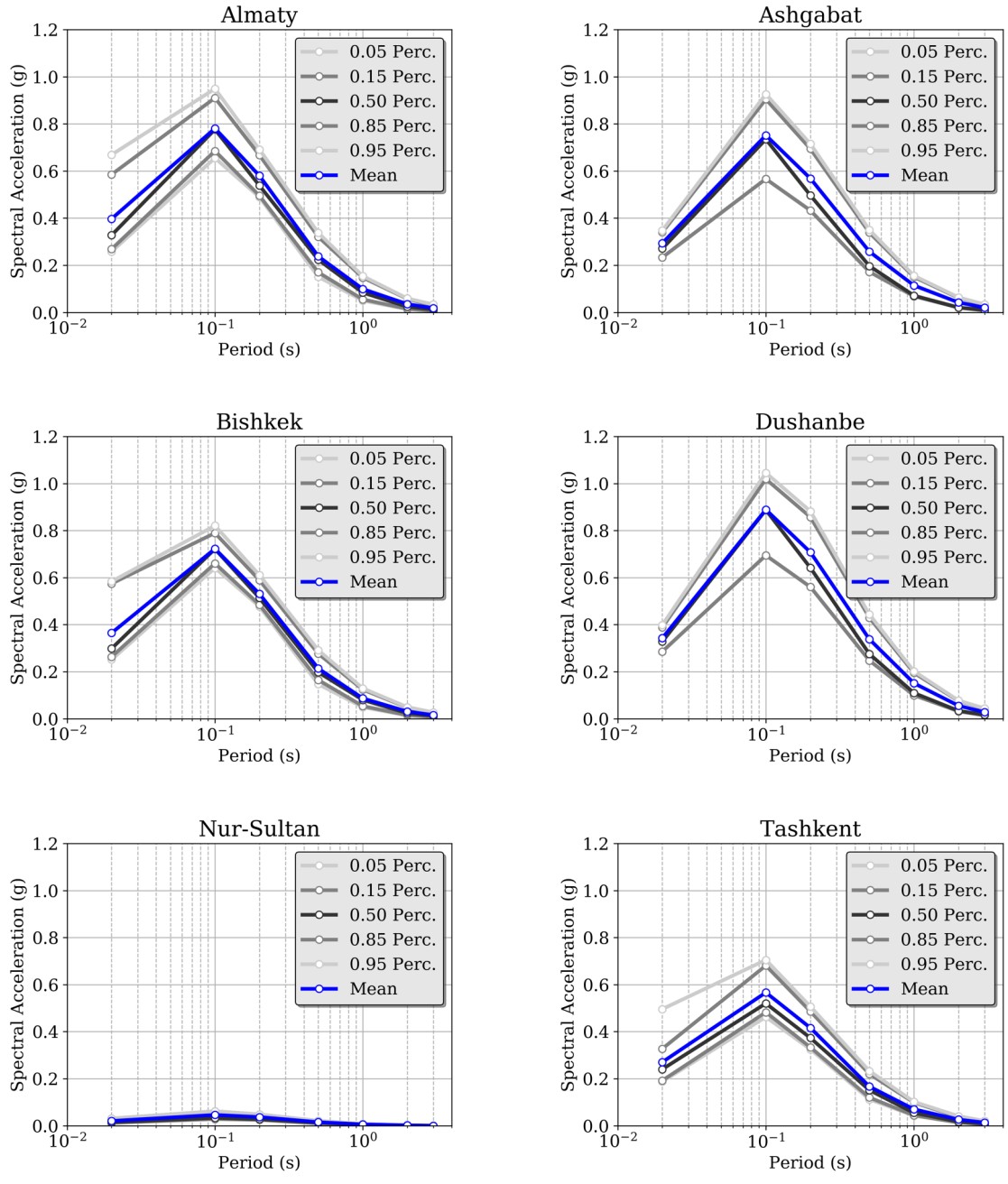

**Figure 15.** Example of uniform hazard spectra (UHS) calculated at six selected target sites (all state capitals plus Almaty, Kazakhstan) for an exceedance probability of 10% in 50 years. Note that the sharp amplitude peak is due to the absence of periods less than 0.1s and should be considered only as a graphical artifact. The PGA is usually plotted with a period of 0.02s (50Hz).

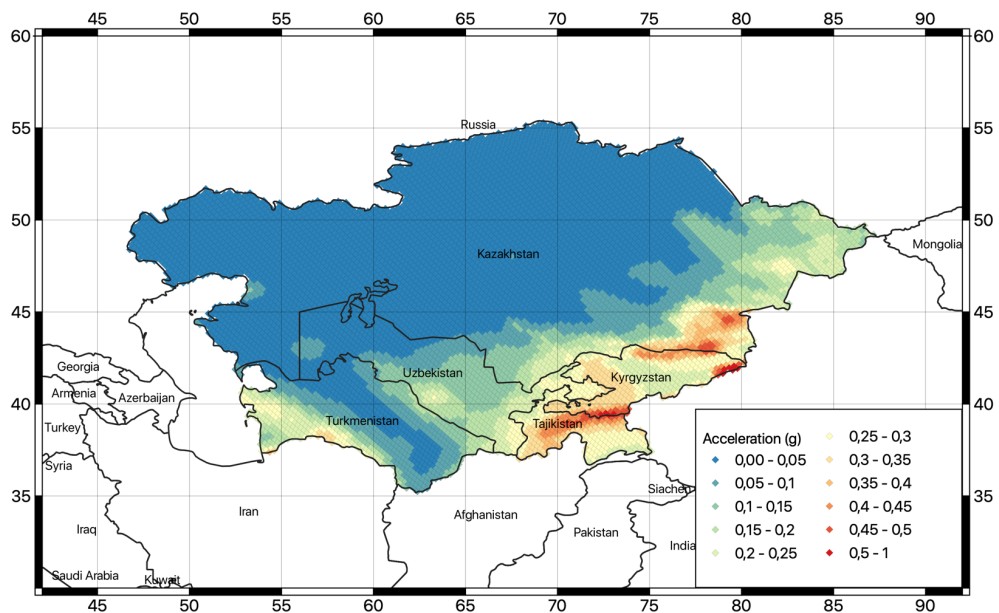

**Figure 16.** Map of calculated peak ground accelerations (PGA) with an exceedance probability of 10% for a study period of 50 years (corresponding to a return period of about 475 years) for rock conditions (Vs30 of 800m/s).

## 10 Discussion

Comparison with previous PSHA studies shows general agreement, although with some noticeable local differences. For example, GSHAP predicts fairly comparable PGA values at 10% PoE for the entire stable continental part (see Sect. S3), whereas peak accelerations are overestimated in the more active southern earthquake belt of Tajikistan and Turkmenistan, with PGAs often exceeding 0.6 g. In the current model, this threshold is exceeded only in a few areas of Tajikistan, while accelerations in Turkmenistan are generally below 0.4 g for this PoE. Consistent results were also found between the current model and more recent calculations by Silacheva et al. (2018) for Kazakhstan and specifically for the city of Almaty, with PGA of around 0.38 g. In Kyrgyzstan, peak accelerations were found in the range of 0.2-0.4 g, which is close to the mean results of Abdrakhmatov et al. (2003). Comparing the hazard curves and uniform hazard spectra of Ischuk et al. (2018) for Almaty, Bishkek, Dushanbe, and Tashkent, slightly larger accelerations (with about 0.1 g difference) are found in the different time periods, although the overall relative response is consistent.

Although not essential for risk assessment, the hazard maps for the different return periods have been converted to macroseismic intensity to facilitate comparison with previous hazard studies and to provide a more accessible presentation of hazard results for non-specialists. In this study, the conversion from PGA to MCS (Mercalli-Cancani-Sieberg) and MSK-64

(Medvedev-Sponheuer-Karnik) intensity scales is performed using the conversion relations developed by Faenza and Michelini (2011):

$$Imcs = 1.68 + 2.58 \log10(PGA(g) * 980.665) \tag{2}$$


and the regional relation from Aptikaev (2012):

$$Imsk = 1.89 + 2.50 \log10(PGA(g) * 980.665) \tag{3}$$

Similarly, we tested the Mercalli Modified Intensity (MMI) equivalence as proposed by Worden et al. (2012) and currently
implemented in the USGS ShakeMap software (Wald et al., 1999). Conversion to other scales can be easily performed if appropriate conversion relationships are available.

It should be noted, however, that the direct conversion of acceleration to intensity is a simplistic approach that should be used with caution, especially when comparing with previous hazard results (e.g., EMCA). A proper hazard assessment using intensity prediction equations (IPE) would be more appropriate along with more granular site response information. This is
not done here because it is not necessary for the risk assessment, which is the ultimate goal of this study. Nonetheless, regionalized IPEs can be implemented and used for direct risk assessment in a possible follow-up to this study.

Conversion of PGA to MCS and MSK intensities yielded almost identical results (see Figure 17 for an example of MKS intensities calculated for a 10% exceedance probability in 50 years). In contrast, the results converted to MMI using the Worden et al. (2012) relationship are systematically lower by about one intensity level (Figure 18). All intensity maps are consistent
with a shear wave reference velocity of 800m/s, in line with guidelines and normative that recommend this shear wave velocity as reference for seismic hazard assessment in engineering applications.

Compared to the earlier results of Ullah et al. (2015) for the EMCA project (see Sect. S3), the MSC/MSK converted intensities of the current study are larger overall by about one intensity unit. However, these differences are likely due to the conversion relation conservatively over-predicting damage even for relatively small PGA values. In addition, it should be noted that Ullah
et al. (2015) performed the intensity calculations directly using IPEs, thus avoiding the uncertainty associated with the additional conversion step. Considering the large uncertainties associated with the macroseismic intensity assessment, the results are nevertheless very comparable and show a fairly consistent spatial pattern between the models. Further similar results are obtained by comparison with the MMI conversion of Worden et al. (2012), confirming the rather large variability associated with direct macroseismic intensity conversion.

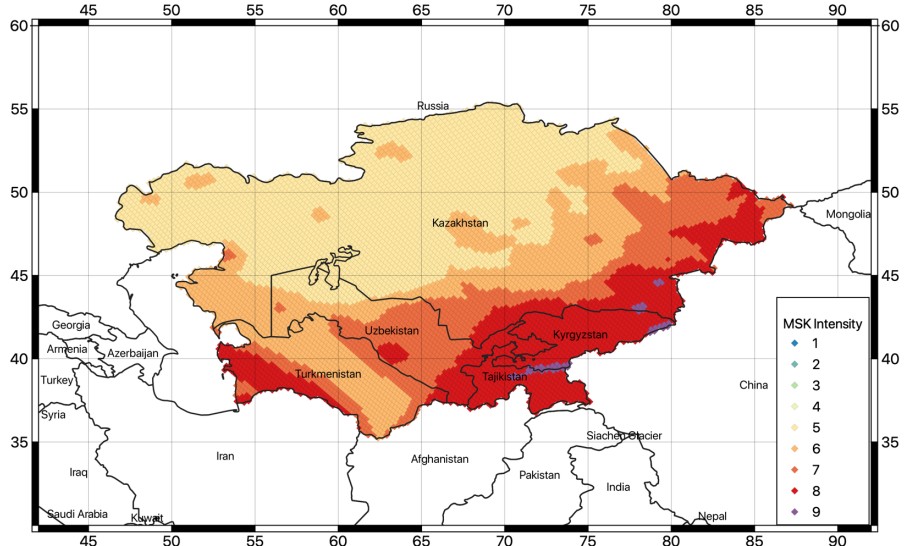

**Figure 17**. Map of PGA converted macroseismic intensity (MSK-64) calculated for this study with an exceedance probability of 10% for a study period of 50 years (corresponding to a return period of about 475 years).


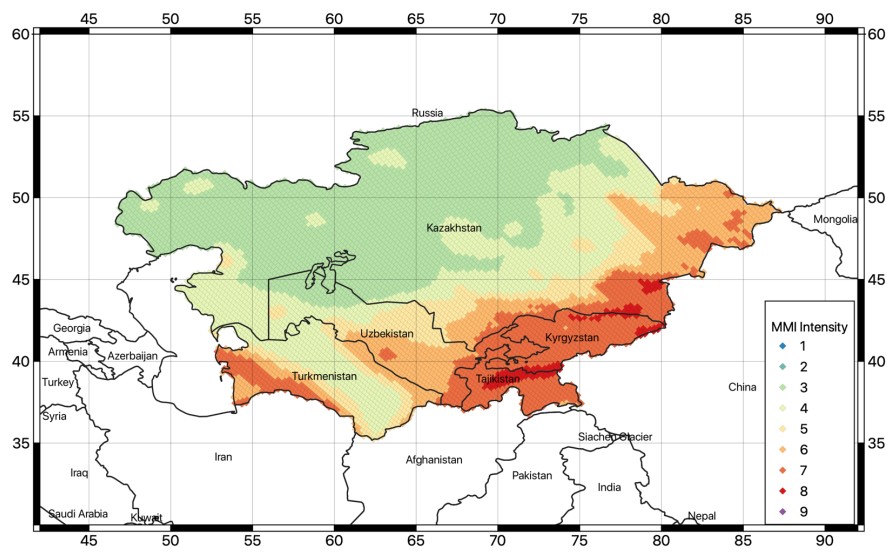

**Figure 18.** Map of PGA converted macroseismic intensity (MMI) calculated for this study with an exceedance probability of 10% for a study period of 50 years (corresponding to a return period of about 475 years).

## 10 Conclusions

This paper proposes a new probabilistic earthquake hazard model for the Central Asian countries of Kazakhstan, the Kyrgyz Republic, Tajikistan, Turkmenistan, and Uzbekistan. The proposed hazard model was developed by the consortium members

in close cooperation with the local scientific partners of the project, whose contribution was also essential for the collection and harmonization of the input data.

In probabilistic seismic hazard analysis, the opinion of experts often plays a crucial role, especially when it comes to controversial or weakly constrained topics. We adopted a structured approach that involved soliciting input and judgement from scientists and professionals with expertise in seismic hazard analysis and engineering, particularly from local communities in the target region. As part of this process, targeted meetings, open discussions and dedicated workshops were held to gather insights and perspectives on various controversial aspects of the study. These aspects included the characterisation of the source model, the assessment of epistemic variability in ground motion prediction and the definition of the logic tree structure. The collected expert opinions were carefully analysed and integrated into the development of the final models and the interpretation of their results.

The major issue affecting the presented model is undoubtedly the shortage of strong-motion recordings within a rupture-to-site distance less than 80km, to be used for selection and validation of existing ground motion prediction models. In this study, the decision on most suitable GMPEs is made primary on the basis of indirect information that relies on a set of tenable assumptions from seismotectonic considerations but, strictly speaking, lacks an empirical validation. The future establishment of new strong-motion stations at potentially hazardous sites and the strengthening of existing seismic networks will be a major step forward in verifying the applicability of existing ground motion prediction models at short distances and in encouraging the development of new locally calibrated models.

Besides that, the definition of more accurate seismic site response model, accounting for the variability of the local geology, is the base to move from regional to site-specific hazard studies, which are essential for a targeted risk analysis. In particular, there is a clear need to incorporate frequency dependent information (e.g., a full assessment of soil response based either on modelling or empirical observations, e.g., Poggi et al., 2014), as opposed to the standard single-term soil proxies (e.g., Vs30, geotechnical classification) that are proven to be too uncertain for the purpose of site-specific hazard analysis. However, except for targeted microzonation studies in major cities, this information is often very limited for the territory. A major investment in this direction would therefore be highly desirable. In addition, the availability of new strong-motion recordings would support site-specific hazard studies that require empirical data for the calibration and verification of numerical seismic-response models. This could be a possible extension of this project in the second phase.

The modelling strategy used in this study is state-of-the-art for probabilistic seismic hazard assessment at the regional scale. However, the current model does not - yet - cover the level of detail typically required for the development of national hazard models (see e.g., Gerstenberger et al., 2019), such as those used to underpin national building codes, although it provides the essential information needed for such an application. Until better studies are conducted, the results of this study can be used to estimate seismic hazards and to promote awareness of seismic hazards in local government institutions. Extension of the present model to the national level and to an urban scenario is clearly a natural follow-up as new local information (e.g., nearby fault studies and site response analyses, records of weak and strong ground motions) becomes available. The overall risk modelling approach used in this study, and in particular the hazard estimates calculated herein, are appropriate for regional

calculation of losses, which is the ultimate goal of this program. The modelling approach adopted is potentially suitable for implementing disaster risk financing applications, such as the development of regional insurance strategies and parametric solutions based on model triggers, as has been done in other regions around the world (e.g., CCRIF for the Caribbean and Central America regions, ARC for the African continent).


## Data availability

All data presented in this paper, including the harmonized earthquake catalog, the active fault database, the PSHA source model files in OpenQuake format and the corresponding calculation results, are available on the World Bank data portal (https://datacatalog.worldbank.org) along with the technical reports produced during the SFRARR project.


## Acknowledgments

The "Strengthening Financial Resilience and Accelerating Risk Reduction in Central Asia" (SFRARR) Program is funded by the European Union, managed by the Global Facility for Disaster Reduction and Recovery (GFDRR) and implemented by the World Bank.

We would like to thank all the project team members, the local partners of the consortium and the World Bank specialists, in particular Dr. Stuart Alexander Fraser and Dr. Madina Nizamitdin, for their constructive contribution to the project.

## Disclaimer

The authors of this document are part of an international consortium of experts that has been hired by the World Bank to implement many of the work under the SFRARR Central Asia program. This model has been produced with the assistance of the European Union, the World Bank, and GFDRR. The sole responsibility of this publication lies with the author and can in no way be taken to reflect the views of three institutions. The European Union, the World Bank, or GFDRR are not responsible for any use that may be made of the information contained therein.


## Author contribution

VP was responsible for coordinating the earthquake hazard component of the SFRARR program.  All co-authors contributed to data collection and review, the model implementation and to the discussion of key results. VP prepared the manuscript with contributions from all co-authors.


## Competing interests

The authors declare that they have no conflict of interest.

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
