# Peer review of "Development of a regional probabilistic seismic hazard model for Central Asia"

_Natural Hazards and Earth System Sciences, 2023_

## Author Comment (AC1)

**Reply to RC1**

We would like to thank the anonymous reviewer for his evaluation of our work, whose suggestions will certainly contribute to improving the quality of our manuscript. We have considered each suggestion thoroughly and have taken steps to implement them in a meaningful way. In this response, we provide detailed explanations and revisions that reflect our efforts to address the reviewer's comments. In addition, where necessary, we have carefully incorporated these improvements into the revised version of the manuscript to ensure that the final document reflects our commitment to excellence and responsiveness to feedback.

1) The number of citations is extremely small with respect to the argument. Seismic hazard assessment is a complex procedure that consider the previous experiences and the many data used for building the model; many affirmations made by the authors without an adequate reference must be referred to the authors themselves, but often this is not the case. In the following some examples.

   *In the new version of the manuscript, we have included new references to existing literature, also in accordance with the following suggestions.*

2) The previous model for this area has been released more then 10 years ago (in 2012 the EMCA project released its model in the frame of the GEM activities). As the authors refer, after 2012 many studies have been realized in the single countries: in the period 2018-2021 all the country involved in this project were studied by several authors, in some cases with applications of the model to the building code. So, it's not clear why a new study is necessary and what are the main criticalities of the previous works, if any. Please, add a comment on this issue.

   *The SFRARR project was led by a consortium of international scientists from various private and public research organisations, including representatives from the Central Asian countries mentioned in this study. Some of these representatives were also involved in the national initiatives mentioned in the introduction. The consortium directly incorporated their experiences into the model presented. The feedback from these national experts influenced many modelling decisions, including the zonation of the sources (six versions were progressively developed and discussed during the project), the homogenisation of the data, the tectonic regionalisation and much more.*
   *Among the benefits of the present study, it should be noted that unlike the national initiatives, which were primarily carried out on an individual basis, the aim of the SFRARR project was to harmonise these contributions into a single regional model. This approach was intended to build on and improve previous efforts, such as the EMCA model published in 2012, by bringing together different expertise and data sources into a comprehensive and harmonised framework.*

3) On row 107, the authors comment that the hybrid approach allows a more realistic representation of the seismicity. In my opinion, in general this is true, but it depends on the practice of design large seismic source area due to the poor knowledge about the seismogenic processes.

*We agree with the reviewer's comment on the general usefulness of the hybrid approach to seismic hazard modelling. This approach is particularly valuable when the delineation of higher resolution source areas proves difficult due to limited seismogenic constraints or, alternatively, when very large regions are considered. In any case, it is fair to point out that accurate zoning can work as well or even better than hybrid modelling under certain conditions, provided that the local process of earthquake generation is sufficiently understood. We have emphasised this consideration more clearly in the manuscript.*

4) Section 3 contains very few information about the definition of the source models. As far as I understood, the definition of seismic source zones (figure 1) is based only on seismic information. There is not any seismotectonic consideration. Is it correct?

*Unfortunately, important details on the development of the model were not included in the manuscript for reasons of space. To counteract this limitation, the most important project results have been divided into two accompanying publications within this special edition by the same team of authors. However, further information can be found in the World Bank's online project report, which is freely accessible on the World Bank's documentation platform.*
*Regarding the definition of the seismic source areas shown in Figure 1, we confirm that our approach has taken into account all available seismotectonic information and has not only analysed the distribution of seismicity. These considerations were discussed in detail in several project workshops with local experts during the construction phase of the source model, where several versions were proposed and iteratively refined.*
*While it is not possible to discuss all seismotectonic considerations in detail in the manuscript, we have added a clarification to address this aspect more explicitly (see also answer to question n. 6).*

5) Row 132: what does it means that this is the accepted version of the source model? From whom? Is it this information useful or necessary?

*We agree with the reviewer that the current definition lacks context and could be misleading. The acceptability of the proposed model version refers to the entire creation process, which aimed to reach a reasoned consensus among the consortium participants. It is important to emphasise that scientific partners in Central Asia and local representatives from different countries actively contributed to both the development and review of the model. Therefore, it was necessary to reach a certain level of consensus considering the different scientific views on certain controversial issues.*

6) Also, the definition of the tectonic groups it's not fully explained. When the authors write that the groups "are assumed to have comparable behavior…" on what basis their judgment is based? Only the earthquake catalog or other data? It is also missing any comment or comparison with previous source model.

*We understand the reviewer's concerns. Unfortunately, due to space constraints, it was not possible to summarise the numerous considerations that led to the development of*

*the zonation model. As briefly highlighted in the manuscript, the grouping was done by combining the analysis of seismicity data with seismotectonic considerations.*

*For example, statistical data on the distribution of focal mechanisms and the empirical magnitude frequency distributions were analysed together with the characteristics of the main active fault systems (presented in the companion paper to this study) in the context of regional tectonic structures and boundaries.*

*To provide a practical example of the construction process, Zone D was found to encompass a tectonic domain that is clearly separated by the stable features of the West Siberian craton (Zone E). As also indicated by the available source mechanisms, Zone D is characterised by a mixed regime, albeit with a dominance of large transpressive fault systems (e.g. Talas-Fergagna fault, Irtysh shear zone) that have influenced the southeastern evolution of the Tianshan Massif (Chen et al. 2022). Towards the south, a change in seismotectonic style becomes evident (Zone G), where the main reverse mechanisms increasingly dominate and large trust systems develop along the suture zone with the former cratonic terrains of the Tarim region (Angiolini et al., 2013). Here, seismic productivity is increasing and large magnitudes have occurred in the past. Further south, a mixed tectonic style is again present (Zone C), while seismicity becomes typical of continental collision (Zone F), with larger and deeper events along the Pamir thrust system (e.g. Murodov 2022). Towards the west, a clear separation between the tectonic styles of the systems at the boundary between the Turan Platform (Zone B) and the Karakum terrains (Zone A) has also been noted along the ideal southwestern extent of the Pamir suture zone (see Ghassemi and Garzanti, 2018 for a comprehensive review).*

*We have now expanded the discussion in the manuscript, although an exhaustive description of the entire argument supporting the construction of the zonation model cannot be included due to the limited length.*

7) About Section 3.2 (Deep seismicity zones), in figure 2 the position of the letters L and H are over the same zone. I understand that the two zones are overlapping, but from the caption I assume that the letter H refers to the wider area with the pale color.

*Thank you for pointing out this inconsistency, which is definitely confusing for the reader. Zone L is indeed the deeper and smaller zone. We have corrected the problem in the new version of the manuscript.*

8) At row 147, I suggest the use of the term "deeper" instead of "less". Even if English is not my mother tongue, as written I understand the deep earthquakes occur at 20 or 30 km.

*The suggestion is well received. We have replaced "less" with "deeper" in the new version of the manuscript.*

9) Section 4.2 (Occurrence rate model). The definition of seismicity rates is crucial in any seismic hazard model and object of many assumptions and operational choices by the modelers. In this field, it is normal to refer to analogue experiences. On the contrary, in this section there is only one reference about the Mmax estimation. I would like ask to the authors what is the approach adopted for the declustering; most used approaches (Gardner & Knopoff, 1974 or Reasemberg & Jones, 1985, among many others) lead to numbers of removed events very different.

*We appreciate the reviewer's comment on the importance of declustering in probabilistic hazard analysis, which we also consider crucial. Although this topic was discussed in detail in section 2.7 of the companion article in the same special issue, focusing on the input datasets compiled for the analysis, we acknowledge that additional clarity is needed on the declustering approach used in our study.*

*In our analysis, we used well-established window-based declustering approaches, including those proposed by Gardner and Knopoff (1974), Uhrhammer (1986), and Grunthal (1985). These methods were chosen because of their suitability for the data set and their widespread use in similar studies. We have now included a clear reference to section 2.7 of the companion paper in the new version of the manuscript to provide readers with further insight into the employed declustering strategy.*

10) The determination of b-value in two steps was adopted in many studies. Anyone to mention?

*Even if we call it two-stage, the strategy of setting a regional b-value for large zones in advance is quite common and has been used several times in research studies and industrial applications. This is usually necessary when the recorded seismicity is not sufficient to perform a more detailed evaluation.*

*Some relevant published examples are from Vilanova and Fonseca, (2007), Ullah et al. (2015), Ghasemi et al. (2020), Ghione et al. (2021). The first author of this study has also applied the same methodology in different regions of the current GEM Global Earthquake Hazard Model, e.g. in the East African Rift (SSA, Poggi et al. 2017), in North Africa (NAF, Poggi et al. 2020) and in Russia/Mongolia (NEA, Pagani et al. 2020).*

11) At row 184 it is reported a sentence that I have to dispute: "It should be additionally noted that the width of the non cumulative magnitude bins is not required to be uniform". In my opinion, based on more than 30 years of expertise, It's the first time that I read something like this. The bin width is a delicate point of the analysis, since it determines the b-value (Marzocchi et al., 2020; doi:10.1093/gji/ggz541). Even more so, the variable width is not acceptable. Let's suppose that in the bin for magnitude 7 +- 0.5, all the events reported in the catalog have magnitude greater than 7: if you use 2 bins (with width 0.5) instead of 1 (with width 1), you will obtain 2 points with the same value in the cumulative curve, and this change the resulting fit. For me the assumption made by the authors it's not acceptable.

*We are not sure that we understand the example given by the reviewer. Indeed, a linear fit using a least squares approach for a single data point cannot be reliably performed due to the imbalance between data points and parameters in the model.*

*From a least squares perspective applied to incremental (non-cumulative) rates, each point that needs to be fitted represents the average frequency in a given magnitude interval. When minimising the squared error over the prediction, the adjustment is calculated for each bin for that specific interval, providing a mean of normalisation. In this way, multiple non-overlapping magnitude intervals can be fitted together, each with its own extension, without loss of generality. Of course, several bins (>2) are required to converge to an unbiased solution.*

*On the contrary, it is important to note that the choice of different bin lengths may affect the robustness of the associated occurrence rates, especially in regions with low seismicity. In general, we would recommend using non-uniform binning, where the intervals become progressively larger with increasing magnitude, e.g. following a logarithmic scheme, to ensure a comparable amount of calibration data for the calculation of rates.*
*Nonetheless, we recognise the reviewer's initial point, which is undoubtedly relevant when maximum likelihood approaches and cumulative distribution functions are considered.*

12) Row 187: It's true that most of rates models start at magnitude 4.5, mainly for completeness reasons. In some cases, we know damaging earthquakes for magnitude 4 or less (as an example in volcanic areas with very shallow hypocenters) Could you quote any papers that affirms what you are saying?

*The reason for the introduction of a lowest truncation in the rate models lies indeed in the need to avoid unnecessary integration steps in the hazard integral. From an engineering perspective, severe damage has been occasionally reported from events with magnitudes less than 4, but for specific cases with high frequency accelerations associated with the effects of site conditions and on highly vulnerable buildings. In most standard cases, however, only light to moderate and non-structural damage is to be expected. When damage levels D4-D5 (severe damage up to collapse) are considered with average exposure and rock conditions, 4.0-4.5 is usually considered a reasonable choice that prevents calculations from being performed that do not directly affect the outcome (in term of impact). Furthermore, magnitudes <4.0 generally do not contribute significantly to the hazard controlling scenario for the most commonly used exceedance probabilities in engineering practise, such as 10% in 50 years.*
*Relevant publications include Bommer and Crowley (2017), Azarbakht (2024), which have now been included in the new version of the manuscript.*

13) I understand that Mmax is based only on the information reported in the catalog, i.e., the maximum observed magnitude. Why was the maximum geological magnitude not considered? One example is contained in Woessner et al., 2015 (doi:10.1007/s10518-015-9795-1). Or do you think that magnitudes larger than the observed events are not possible?

*We are of the opinion that events that are greater than those observed can certainly be expected. For this reason, when defining the Mmax of each zone in our model, we always take into account a conservative premium on the maximum observed magnitude (and also consider its uncertainty). In addition, we have considered the epistemic variability of Mmax in the log-tree of the source model.*
*The direct use of geological constraints must be carefully considered. For studies focussing on specific known structures, we would agree to consider the maximum extent of the rupture. However, in the present case, the mapped seismogenic structures are generally not constrained in such detail. Individual fault lines could represent one or more complex systems and information on the actual segmentation is generally lacking, which could lead to a dramatic overestimation of the expected maximum magnitude. For example, when using the AFEAD dataset, many mapped faults may yield unphysical magnitudes if scaling*

*relationships are applied to their entire extent to convert the rupture area to a moment magnitude, as was done in SHARE's FSBG model.*

14) Row 270: the smearing effect due to the adoption of seismic areas depends on the approach adopted to design the areas: smaller are the areas and more the hazard is concentrated on the epicentral areas. The design of areas should contain a sort of "prediction" for those zones with poor knowledge about the historical seismicity.

    *We agree with the reviewer's comment, which also agrees well with our answer to question 3.*

15) Figure 10: I wonder why the tectonic regions are different with respect to the groups of figure 1. As an example, source zone 5 in figure 1 has a different classification in figure 10 if I consider the other zones of group A. There is an explanation?

    *Yes, the difference in grouping between the zonation of the source model and the tectonic regionalisation is due to the fact that they represent different aspects of the earthquake phenomenon. The source zone grouping was based on similarities in the process of earthquake generation at the source level, while the grouping for the tectonic regionalisation type (TRT) was based on the expected differences in the general attenuation behaviour along the path from the source to the site and aimed to differentiate the ground motion prediction equations (GMPEs). Although there are some similarities between the two classification schemes, they are not equivalent as they are based on different assumptions and constraints.*
    *We have clarified this aspect in the new version of the manuscript to provide a better understanding of the reasons for the different groupings in Figures 1 and 10.*

16) Row 371: what are the considerations that allow you to say that in stable continental crust zone an intermediate behavior between active shallow crust and stable crust is expected? I don't say that it is not true, but I would like that you support this sentence with a reference or your comment.

    *The assertion regarding the expected intermediate behaviour in stable continental crustal zones was based on operational considerations of the authors. We came to the conclusion that a purely cratonic attenuation behaviour is unlikely to be expected for regions such as the Kazakh Shield. Assuming that tectonic regionalisation type 2 (TRT2) exhibits less extreme attenuation behaviour compared to standard regions with active shallow crust, it can therefore be inferred that buffer tectonic regions surrounding large active structural systems such as the Tianshan Massif and Turkmenistan would behave in an intermediate manner. This operational choice is partially supported by the associated seismicity patterns, as shown in Figure 4 of the accompanying paper, but can only be verified if sufficient ground motion records are available for comparison with the selected ground motion models.*

17) Section 8. I don't find any description in the manuscript about the 3 options for the assignation of b-value (b, b+0.5, b-0.5). With regard to Mmax, on contrary, in the manuscript I found only a sentence about the branch with Mmax+0.1. I think that the

whole logic tree has to be described together with the strategies adopted for assigning weights.

*We take note of the reviewer's suggestion. In the new version of the manuscript, we have included additional explanations of the strategies for implementing the logic tree.*

18) Row 415: this is a clarification. The results of the calculation are only the hazard curves (not only in OpenQuake engine). Maps and UHS are possible representations!

*We fully agree with the reviewer's clarification. While the hazard curves are indeed the primary results of the calculation, it is important to point out that maps and Uniform Hazard Spectra (UHS) are derived products obtained from the hazard curves representing certain exceedance probabilities and observation times.*
*In the manuscript, we only wanted to emphasise the output product that we made readily available to the reader in OpenQuake format. These data can indeed be downloaded from the World Bank's data portal and will also be included in the journal's repository.*

19) Section 10. I don't understand why the presentation of the model expressed in terms of macroseismic intensity is in this section and it's not in ad hoc section.

*This decision results from the editorial decision to summarise the information from the original project report in the form of an article. In the original report, a separate section was indeed devoted to the discussion of macroseismic intensity, as expected by the reviewer. However, in order to reduce the overall length of the manuscript, we decided to integrate this alternative presentation directly into the discussion section. This approach allowed us to integrate the comparison with existing models in a natural way and to facilitate the corresponding discussion.*

20) Row 434: "Comparison with previous PSHA studies shows general agreement". I don't see the comparison! In the Supplement it is reported the map of GSHAP, released 25 years ago. Probably this is not the best test... In my opinion the comparison has to be performed with EMCA project (most recent study for the same area) or with recent national projects. Not only: I expect a quantitative comparison, not only a comparison of two figures.

*Dear Reviewer, we have reproduced the maps of GSHAP (for PGA) and EMCA (MSK intensity) in Section 3 of the Appendix (see Supplementary Material) in Figure S4 and S5, respectively. The two maps were created using the same colour scale, sample, metrics and extension of the hazard maps from the present study to facilitate quantitative comparison for the reader.*
*We intentionally did not include a map of differences as this would have overemphasised the significance of the discrepancies. It is important to clarify that our aim in comparing these models was not to assert the superiority of one over the other. Instead, we have sought to understand the differences and similarities between them, recognising that all models, including ours, have their own limitations. As George E. P. Box famously said, "All models are wrong, but some are useful." Our comparison therefore aimed to emphasise the usefulness of each model in different contexts, rather than ranking them.*

*As for the GSHAP, we recognise its age and the limitations associated with using this model as a benchmark. However, it is worth noting that GSHAP is still considered in many technical studies. During project development, we were repeatedly asked by partners and reviewers to perform such a comparison, which is why we included it.*

21) Regarding the intensity maps, at row 464 you write: "All intensity maps are consistent with a shear wave reference velocity of 800m/s". This is a strong statement and I ask you to cite a paper or discuss it. Most localities are built near rivers for access to water; this means soil conditions other than rocky ones.

*The use of a shear wave reference velocity of 800 m/s is a common practise in seismic hazard assessment and is often used as a standard for comparison purposes. It provides a standardised basis for the evaluation of seismic hazard in different regions. Furthermore, this choice is in line with the guidelines that recommend 800 m/s as the reference shear wave velocity for seismic hazard assessment in engineering applications. Thus, the use of a standard reference rock is a common abstraction and does not necessarily reflect actual site-specific conditions. While site-specific models could provide more accurate assessments, their applicability may be limited if they are based on the assumptions of regional models, especially in regions where soil conditions vary widely, such as near rivers.*
*Nevertheless, in our study we also calculated a site-specific model, which is discussed and provided in another article by Salgado et al. in the same special issue (presently under review). This model provides a more detailed assessment of seismic risk for the Central Asian countries and takes into account site-specific soil conditions and local geological features.*
*Overall, while we recognise the importance of considering site-specific conditions, the use of a standard reference rock in our intensity maps allows for consistency and comparability with existing studies and provides a basis for further analysis and interpretation.*

22) In the conclusions, again, very few reference, but paper by Poggi et al.. When you talk of the strategies for assess seismic hazard at national scale, for example, you could quote Gerstenberger et al., 2020 (doi:10.1029/2019RG000653). For the international project, also, the references for CCRIF and ARC projects are missing.

*We appreciate the reviewer's suggestion to include additional references in the conclusions section. We believe that these additional references will enrich the discussion and provide readers with further resources to critically analyse the results of this comprehensive project.*

---

## Author Comment (AC2)

**Reply to RC2**

We extend our gratitude to the anonymous reviewer for her/his positive evaluation of our work. We have carefully considered all the suggestions provided and have addressed them comprehensively in this response. Furthermore, we have also incorporated these suggestions into the updated version of the manuscript.

1)  Line 95: What does the word "size" mean? Refers to the dimensions, proportions, or magnitude.

    *We agree that the term 'size' could be misleading when applied to earthquake sources. Since the focus is on geometrical properties, we have replaced 'size' with 'dimension', as suggested.*

2)  Part 3: Is the division of the region into 7 groups and 61 regions based on the current work or previous works? If it has been done in this work, the method needs to be explained.

    *We understand the reviewer's concerns. Unfortunately, due to space constraints, it was not possible to summarise the numerous considerations that led to the development of the zonation model. As briefly highlighted in the manuscript, the grouping was done by combining the analysis of seismicity data with seismotectonic considerations.*
    *For example, statistical data on the distribution of focal mechanisms and the empirical magnitude frequency distributions were analysed together with the characteristics of the main active fault systems (presented in the companion paper to this study) in the context of regional tectonic structures and boundaries.*
    *To provide a practical example of the construction process, Zone D was found to encompass a tectonic domain that is clearly separated by the stable features of the West Siberian craton (Zone E). As also indicated by the available source mechanisms, Zone D is characterised by a mixed regime, albeit with a dominance of large transpressive fault systems (e.g. Talas-Fergagna fault, Irtysh shear zone) that have influenced the southeastern evolution of the Tianshan Massif (Chen et al. 2022). Towards the south, a change in seismotectonic style becomes evident (Zone G), where the main reverse mechanisms increasingly dominate and large trust systems develop along the suture zone with the former cratonic terrains of the Tarim region (Angiolini et al., 2013). Here, seismic productivity is increasing and large magnitudes have occurred in the past. Further south, a mixed tectonic style is again present (Zone C), while seismicity becomes typical of continental collision (Zone F), with larger and deeper events along the Pamir thrust system (e.g. Murodov 2022). Towards the west, a clear separation between the tectonic styles of the systems at the boundary between the Turan Platform (Zone B) and the Karakum terrains (Zone A) has also been noted along the ideal southwestern extent of the Pamir suture zone (see Ghassemi and Garzanti, 2018 for a comprehensive review).*
    *We have now expanded the discussion in the manuscript, although an exhaustive description of the entire argument supporting the construction of the zonation model cannot be included due to the limited length.*

3) Line 146. How is the depth uncertainty included?

*The depth information comes directly from the solutions of the seismological agencies used to compile the homogenised catalogue (more detailed in the article accompanying this article in the same special issue). Unfortunately, the input data often lack the uncertainties related to each individual solution. Nevertheless, the statistical analysis performed in Section 4.1 ("Hypocentral depth distribution") helps us to build the probabilistic source depth distribution model.*
*In fact, OpenQuake accepts a probability density distribution of the hypocentral depth for each area source, which we then derive from the observations by regularising the depth histogram. Such a distribution is also used to delineate the depth boundaries of the source zones.*

4) Line 151. H and L areas are not distinguishable in the map (Figure 2). In this case, as mentioned in the previous question, the discussion of depth uncertainty needs to be included in the analysis and text.

*Thank you for pointing out this inconsistency, which is definitely confusing for the reader. Zone L is the lower and smaller zone. We have corrected the problem in the new version of the manuscript.*

5) In Figure 3, the H and L zones are not consistent with the text. For example, is the depth of 150 the limit or the depth of 170?

*Thank you for spotting the inconsistency. The reviewer is correct. The depth limit was set at 170 km, which is consistent with the depth distribution contained in the model. 150 km is a holdover from an older version of the initial model, the description of which was inadvertently not updated in the manuscript. The correct limit of 170 km has now been inserted.*

6) Figure 4. The Gutenberg-Richter (in its logarithmic form) is a linear relationship; why is the fitted curve non-linear?

*In contrast to the original (unbounded) formulation, which indeed exhibited linearity, the truncated version of the Gutenberg-Richter relationship (as shown in Equation 1) introduces a dependence on the maximum magnitude parameter (Mmax). This parameter is used to restrict the occurrence of events that exceed this magnitude and effectively exclude them from the cumulative distribution. As the magnitude approaches Mmax, the truncated relationship tends asymptotically towards zero, making its representation inherently non-linear.*
*The introduction of the truncated Gutenberg-Richter model was historically aimed at preventing the occurrence of 'unphysical' magnitudes (although the definition of what is 'unphysical' remains the subject of ongoing debate). The original Gutenberg-Richter model, since it was not truncated, could theoretically generate magnitudes of arbitrary values, albeit with extremely low probabilities (e.g. M=10).*

7) Line 170. Why the (one-side) truncated Gutenberg-Richter relation is used? Why the mmin is not included? In the rest of the text, contradictions can be seen in this field and the double truncated Gutenberg-Richter relationship is used.

*The truncated Gutenberg-Richter (G-R) relation is cumulative in exceedance (representing the number of events with magnitude greater than "m") and does not require the inclusion of Mmin as a parameter in its formulation. The lowest truncation, usually labelled Mmin, is not an intrinsic parameter of the distribution itself and can be set arbitrarily without affecting the generality of the model. It is important to emphasise that the inclusion of the lowest truncation does not mean that there are no low magnitude events, but that events unlikely to cause significant damage to the target structures are excluded from the calculation of the hazard integral. Therefore, a generally accepted value for Mmin, e.g. 4.5 (or occasionally 4 in more conservative cases), is considered appropriate for most engineering applications. The importance of Mmin arises primarily when the truncated G-R is converted into a probability density function (PDF) for solving the hazard integral.*

8) Line 172. Is Gutenberg-Richter's relation applicable for values lower than completeness magnitude?

*Yes, the Gutenberg-Richter relation (G-R) remains applicable for magnitudes below the completeness magnitude, provided that the assumption of occurrence according to the G-R relation is correct. Completeness magnitude, often referred to as incompleteness, does not refer to the inherent nature of the magnitude-frequency distribution itself, but to the seismic catalogue used to calibrate the occurrence model. Events below a certain magnitude threshold may be inadequately represented in seismic catalogues, e.g. due to limitations in the coverage or sensitivity of the seismic network or due to other reporting limitations.*
*However, it is important to recognise that these events exist in nature. If the G-R relationship is appropriately calibrated, it can accurately predict their occurrence even if they are not fully captured in the calibration dataset.*
*Another problem arises when the seismic catalogue is incomplete at the upper end, i.e. when the maximum possible magnitude is not included in the historical records, e.g. due to very long return periods that exceed the duration of the available data set. In such cases, a conservative approach is to estimate an upper limit for the largest observed magnitude, taking into account the possibility of generating these larger events. This adjustment can have a significant impact on the truncated G-R relationship, as shown in Equation 1.*

9) Line 177. Why is the list square method used? Is the data homogeneous? Considering the age of the countries in the region and the long history, are historical data included or not?

*Least squares (LS) and maximum likelihood (ML) are two widely used methods for calibrating the G-R occurrence model. The choice between LS and ML for calibrating the G-R model requires careful consideration of their respective advantages and limitations. While the ML method is often favoured for its ability to fit statistical distributions such as cumulative functions, it may tend to over-fit the lower magnitude range, leading to an underestimation of larger magnitudes, especially in regions with sparse data or low*

*seismicity. In contrast, LS fitting offers greater robustness and is less prone to overfitting, making it particularly suitable for regions with low seismicity or short catalogues. On the other hand, it suffers the impracticality to be performed on incremental (non-cumulative) magnitude bins to avoid data dependence. The authors have found that LS adjustment provides better results in such regional scenarios, as evidenced by its successful application in other challenging regions, such as Africa.*

*Regarding the inclusion of historical data, it is worth mentioning that these data are indeed included in the calibration of the occurrence model. Historical seismic records spanning long periods of time are essential to constrain the long-term seismic activity in the region.*

*The homogeneity of data across the region is essential for a robust seismic hazard analysis. As part of this project, efforts were made to harmonise and standardise seismic datasets from different countries in Central Asia, taking into account differences in seismic monitoring infrastructure and data recording practises. A detailed description of the homogenised catalogue used in this study can be found in a companion paper in the same special issue, written by the same team (presently under review)*

10) Line 187. The minimum magnitude of the Gutenberg-Richter of 4.5 does not match the one- side truncated Gutenberg-Richter relation. There seems to be a problem in the sentence " rate of earthquakes with magnitude greater than 0 "; because minimum magnitude of the Gutenberg Richter's has nothing to do with the potential of failure, but with the completeness magnitude. Therefore, the sentence seems to require revision.

*As already mentioned in this answer, it is important to distinguish between the minimum magnitude parameter (Mmin), which is used in the calculation of the hazard integral and is often selected based on the damage potential of seismic events, and the completeness magnitude, which denotes the lowest magnitude level that is fully represented in the data set. The phrase 'rate of earthquakes with magnitude greater than 0' refers to the productivity parameter of the Gutenberg-Richter relationship, commonly known as the a-value. Although it usually refers to magnitude 0 (the intercept), there are cases in the literature where the a-value is associated with other magnitude levels. This difference in terminology can lead to occasional discrepancies in interpretation, but does not affect the basic principles of seismic hazard analysis.*

11) In Table 2, the weights are assigned on what basis?

*The weights assigned to the rupture mechanisms of each source group in Table 2 were determined by comparing the moment tensor solutions, analysed using Kaverina's classification diagrams (e.g. Figure 6), with the distribution of the predominant fault systems in the regions (detailed in the authors' companion paper in this special issue).*

*In Group D, for example, there are two main mechanisms, reverse and strike-slip, with a similar number of reported solutions, leading to an initial probability fractionation of 50-50%. However, the definition of the actual strike direction from the mapped faults was ambiguous, leading to the identification of two main families of orientations. As there was no evidence for the dominance of one family over the other, we further split the original 50% probability into 25% to 25%. Similar considerations were made for the other groups.*

12) Table 3, what is the source (reference) of the table?

*The values in Table 3 are strictly derived from the depth limits defined for the source zones and thus from the seismicity analysis performed in Section 4.1. To define the LSD and USD boundaries, we have in practise allowed the ruptures occurring at the interface between the different depth zones to extend to a certain limit, which is between 15 and 30 km depending on the expected magnitudes. It should be noted that LSD and USD may not be exact values, but conservative limits to avoid the development of ruptures with unrealistic depth extent.*

13) Line 304: Seismic coefficient = 0.1, why?

*By definition, the aseismic coefficient represents the fraction of the total moment accumulation rate that is not released by earthquakes (i.e. aseismic). The calibration of this parameter is a challenge as there is no consensus within the seismological and geodetic communities on its optimal value and direct methods for its evaluation are currently lacking. Furthermore, the literature on this topic is quite limited.*
*It would be unrealistic to set the aseismic coefficient to 0, as part of slip is inevitably released by plastic deformation. Conversely, values greater than 0.2 often lead to inconsistencies between the total moment derived from slip rate and that observed from seismic events in the catalogue.*
*Through comparative analyses of the hazard levels derived from fault models and seismic catalogues, we evaluated different values for the seismic coefficient. A value around 0.1 was found to be workable as it balances the need for realism with the limitations imposed by the available data and modelling techniques. Although this value is a rough estimate, it agrees reasonably well with the empirical observations and ensures consistency between the modelled hazard levels and the observed seismic activity.*

14) In the manuscript, the experts' opinion is repeatedly mentioned. What was the mechanism of use and criteria?

*In probabilistic seismic hazard analysis, expert opinion often plays a crucial role in modelling, especially when controversial or weakly constrained topics are involved. We have adopted a structured approach that involves soliciting input and assessments from scientists and professionals with expertise in seismic hazard analysis and engineering, particularly from local communities in the target region (partners of the present consortium).*
*The mechanism for incorporating expert opinion includes holding various targeted meetings, open discussions and special workshops to gather insights and perspectives on various aspects of the study that were considered 'dubious" or contentious. These aspects include characterising the source model, assessing epistemic variability in ground motion prediction and defining the logic tree structure. Once the expert opinions are collected, they are carefully evaluated and synthesised.*
*This process involved identifying areas of agreement among the experts, addressing conflicting opinions through further discussion and analysis, and finally integrating the experts' insights into the development of the final models and the interpretation of their results.*

*Overall, the inclusion of expert opinions significantly improved the robustness and validity of our research findings and provided valuable perspectives from practitioners and researchers who were actively involved in the study. Several of these activities were organised as part of this project for the different components of the multi-risk model.*

15) In Tables 5, 6 and figure 12, what is the method for assigning weights?

*If sufficient calibration data, such as records of strong motion recordings, are available, weights for ground motion models (GMPEs) can be assigned based on their degree of agreement with observed ground motions. Efficient ranking methods, such as the LLH approach proposed by Scherbaum et al. (2009), are commonly used for this purpose. However, in our case, the available records for the region were not sufficient to perform a robust data-driven evaluation. Therefore, a more conservative approach to assigning weights was required.*

*In Tables 5 and 6, two equally weighted GMPE models were selected for the shallow tectonic conditions (Active Shallow and Stable Continental), while only one suitable GMPE was identified for Deep Seismicity. However, the use of a two-level tectonic zonation (as shown in Table 6) allowed us to also consider intermediate tectonic conditions, such as the TRT2, where all four GMPEs of Active Shallow and Stable Continental are considered.*

*The reason for maintaining the logical separation between the two steps is to facilitate future changes to the weighting scheme as new data becomes available. This approach ensures flexibility in adjusting the weighting scheme to incorporate additional calibration data or to refine the selection of GMPEs based on improved understanding or progress in the field.*

16) In Table 6, most of the weights are 0 and 1, and it doesn't seem that the two-step method (line 380) has much effect on the results.

*In the specific regional context considered in our study, the two-level tectonic zonation becomes particularly relevant when the hybrid buffer region between the active shallow (AS) and the stable continental (SC) tectonic environments is introduced. This hybrid region allows the blending of ground motion models, effectively accounting for intermediate tectonic conditions. While the same goal could have been achieved by directly assigning a weight of 0.25 to each of the four models in a one-level zonation, a two-level zonation provides a level of abstraction to better deal with mixed regions.*

*We acknowledge that in the context of our study, with its relatively simple regional ground motion model, two-level zonation may not be strictly necessary. However, we believe that this study serves as an illustrative use case for the methodology and demonstrates its potential utility in more complex scenarios where different mixed environments are present. By applying the two-step approach, we provide a framework that offers flexibility and scalability, ensures adaptation to varying degrees of tectonic complexity, and facilitates future refinements as additional data and insights become available.*

17) How does Figure 11 help present the paper?

*The Trellis diagram shown in Figure 11 is a valuable tool for evaluating the performance of selected ground motion prediction equations (GMPEs) for different types of ground*

*motion intensity measurements, magnitudes, and distances relevant to the analysis. These plots provide a comprehensive visualisation of the performance of each GMPE under different conditions and facilitate the process of model selection, especially in situations where direct ground motion observations are limited.*

*By comparing the performance of different GMPEs with the corresponding hazard results from different branches of the logic tree, it also becomes easier to recognise the specific contributions of each model to the overall hazard assessment without the need for disaggregation analysis.*

*In addition to comparing the mean ground motion estimates shown in the figure, it is worth noting that the variability of the overall standard deviation (not shown in the manuscript for conciseness) is also generally taken into account. This comprehensive evaluation provides a more thorough understanding of the uncertainties associated with ground motion predictions and their implications for seismic hazard analysis.*

---

## Referee Report (RR1)

Dear Professor Filippos Vallianatos

I respectfully acknowledge that the detailed responses provided by the esteemed authors have addressed the my points to a satisfactory level. However, the discussion can continue on two questions"

**7. Line 170. Why the (one-side) truncated Gutenberg-Richter relation is used? Why the $m_{min}$ is not included? In the rest of the text, contradictions can be seen in this field and the double truncated Gutenberg-Richter relationship is used.**

The esteemed authors have had a very appropriate discussion, but the ambiguity has not been resolved.

**8. Is Gutenberg-Richter's relationship applicable for values lower than completeness magnitude?**

The answer of the respected author was "yes", which was ambiguous in my opinion.

Considering these issues, the manuscript is very valuable, and in my opinion, it is the same as its predecessor, and it does not need to be re-examined by me.

Sincerely yours

 Sasan Motaghed

---

## Author Response (AR2)

Dear Editor, dear Reviewer

In response to your request, we have further clarified the two highlighted issues in this revised version of the manuscript. Additionally, we have updated the author affiliations, as already requested by the editorial team in the published companion paper of this article (Part A)

7) Line 170. Why the (one-side) truncated Gutenberg-Richter relation is used? Why the mmin is not included? In the rest of the text, contradictions can be seen in this field and the double truncated Gutenberg-Richter relationship is used.

*We have expanded the text to clarify the issue of representing Mmin in the Gutenberg-Richter (GR) relationship, addressing the reasons for using the one-sided truncated model in certain cases and explaining how Mmin is incorporated in other parts of the analysis where the double-truncated GR relationship is applied.*

*"A lower magnitude cutoff (Mmin) is introduced solely when applying this relationship in the hazard integral, which will be discussed later. Since the relationship is cumulative with respect to increasing magnitudes, this lower truncation does not affect the formulation in Equation (1) or its calibration."*

8) Line 172. Is Gutenberg-Richter's relation applicable for values lower than completeness magnitude?

*We have revised the text to better explain the possibility of extending the Gutenberg-Richter (GR) relationship to magnitudes below the completeness threshold, addressing its general applicability in this context.*

*"It is important to note that the general validity of the GR relationship is often assumed to extend also to magnitudes below the completeness magnitude, which merely defines the data range used for calibrating the relationship's coefficients, without restricting the overall applicability of the formulation."*